# Unifying Width-Reduced Methods for Quasi-Self-Concordant Optimization

**Deeksha Adil**
University of Toronto
deeksha@cs.toronto.edu

**Brian Bullins**
TTI Chicago
bbullins.ttic.edu

**Sushant Sachdeva**
University of Toronto
sachdeva@cs.toronto.edu

## Abstract

We provide several algorithms for constrained optimization of a large class of convex problems, including softmax, $\ell_p$ regression, and logistic regression. Central to our approach is the notion of width reduction, a technique which has proven immensely useful in the context of maximum flow [Christiano et al., STOC'11] and, more recently, $\ell_p$ regression [Adil et al., SODA'19], in terms of improving the iteration complexity from $O(m^{1/2})$ to $\tilde{O}(m^{1/3})$, where $m$ is the number of rows of the design matrix, and where each iteration amounts to a linear system solve. However, a considerable drawback is that these methods require both problem-specific potentials and individually tailored analyses.

As our main contribution, we initiate a new direction of study by presenting the first *unified* approach to achieving $m^{1/3}$-type rates. Notably, our method goes beyond these previously considered problems to more broadly capture *quasi-self-concordant* losses, a class which has recently generated much interest and includes the well-studied problem of logistic regression, among others. In order to do so, we develop a unified width reduction method for carefully handling these losses based on a more general set of potentials. Additionally, we directly achieve $m^{1/3}$-type rates in the constrained setting without the need for any explicit acceleration schemes, thus naturally complementing recent work based on a ball-oracle approach [Carmon et al., NeurIPS'20].

## 1 Introduction

We study a class of constrained optimization problems of the following form:

$$\min_{\boldsymbol{Ax}=\boldsymbol{b}} \sum_i \boldsymbol{f}\big((\boldsymbol{Px})_i\big) \tag{1}$$

for convex $\boldsymbol{f} : \mathbb{R} \to \mathbb{R}$, where $\boldsymbol{A} \in \mathbb{R}^{d \times n}$, $\boldsymbol{b} \in \mathbb{R}^d$, $\boldsymbol{P} \in \mathbb{R}^{m \times n}$, with $d \leq n \leq m$. Specifically, we are interested in the case where $\boldsymbol{f}$ satisfies a certain higher-order smoothness-like condition known as $M$-quasi-self-concordance (q.s.c.), i.e., $|\boldsymbol{f}'''(\boldsymbol{x})| \leq M\boldsymbol{f}''(\boldsymbol{x})$ for all $\boldsymbol{x} \in \mathbb{R}$. Several problems of significant interest in machine learning and numerical methods meet this condition, including logistic regression [Bac10, KSJ18], as well as softmax (often used to approximate $\ell_\infty$ regression) [Nes05, CKM+11, EV19, Bul20] and (regularized) $\ell_p$ regression [BCLL18, AKPS19].

A very useful optimization technique, first introduced by [CKM+11] for faster approximate maximum flow and later by [CMMP13] for regression, is that of width reduction, whereby they used it to improve the iteration complexity dependence on $m$, the number of rows of the design matrix from $O(m^{1/2})$ to $\tilde{O}(m^{1/3})$, and where each iteration requires a linear system solve. Later work by [AKPS19] for high-accuracy $\ell_p$ regression, building on an $O(m^{1/2})$-iteration result from [BCLL18], again showed

35th Conference on Neural Information Processing Systems (NeurIPS 2021).

how width reduction could lead to improved $\tilde{O}(m^{1/3})$-iteration algorithms. As a drawback, however, these approaches rely on potential methods and analyses specifically tailored to each problem.

Building on these results, we present the first *unified* approach to achieving $m^{1/3}$-type rates, at the heart of which lies a more general width reduction scheme. Notably, our method goes beyond these previously considered problems to capture *quasi-self-concordant* losses, thereby further including well-studied problems such as logistic regression, among others. By doing so, we directly achieve $m^{1/3}$-type rates in the constrained setting without relying on explicit acceleration schemes [MS13], thus complementing recent work based on a ball-oracle approach [CJJ$^+$20]. We additionally note that, given the ways in which our results achieve improvements similar to those of [CJJ$^+$20], we believe our work hints at a deeper, though to our knowledge not yet fully understood, connection between the techniques of width reduction and Monteiro-Svaiter acceleration.

## 1.1 Main Results and Applications

We first present in Section 3 a width-reduced method for obtaining a crude approximation to (1) for quasi-self-concordant $\boldsymbol{f}$. At a high level, our algorithm returns an approximate solution $\tilde{\boldsymbol{x}}$ that both satisfies the linear constraints and is bounded in $\ell_\infty$-norm by $O(R)$, where $R$ is a bound on the norm of the optimal solution. Following from Theorem 3.3, the result below shows how, for the problem of minimizing softmax (parameterized by $\nu > 0$), i.e., $\mathrm{smax}_\nu(\boldsymbol{Px}) = \nu \log\left(\sum_i e^{\frac{(\boldsymbol{Px})_i}{\nu}}\right)$, we can bound the norm of the solution by $(1+\nu)R$.

**Theorem 1.1.** *Let $\boldsymbol{x}^\star$ denote the optimum of $\min_{\boldsymbol{Ax}=\boldsymbol{b}} \mathrm{smax}_\nu(\boldsymbol{Px})$. Algorithm 1 when applied to the function $\boldsymbol{f}(\boldsymbol{Px}) = \sum_i e^{\frac{(\boldsymbol{Px})_i}{\nu}}$ with $\epsilon = \nu$, returns $\widetilde{\boldsymbol{x}}$ such that $\boldsymbol{A}\widetilde{\boldsymbol{x}} = \boldsymbol{b}$, and*

$$\mathrm{smax}_\nu(\boldsymbol{P}\widetilde{\boldsymbol{x}}) \le (1 + \widetilde{O}(\nu))\mathrm{smax}_\nu(\boldsymbol{Px}^\star),$$

*in at most $\widetilde{O}(m^{1/3}\nu^{-5/3})$ calls to a linear system solver.*

As a consequence of Theorem 1.1 when taking $\nu = \Omega\left(\epsilon/\log^{O(1)}(m)\right)$, we have by Theorem 5.2 a $(1 + \epsilon)$ approximate solution to the problem of $\ell_\infty$ regression with $\tilde{O}(m^{1/3}\epsilon^{-5/3})$ calls to a linear system solver.

Further, we show the following result which can use the approximate solution returned by Theorem 1.1 as an initial point for achieving a high-accuracy solution. We also present in Appendix A a natural extension of our results to minimizing general-self-concordant (g.s.c.) functions.

**Theorem 1.2.** *For $M$-q.s.c. $\boldsymbol{f}$, $\epsilon > 0$, and $\boldsymbol{x}^{(0)}$ such that $\boldsymbol{Ax}^{(0)} = \boldsymbol{b}$ and $\|\boldsymbol{x}^{(0)}\|_\infty \le R$, Algorithm 2 finds $\widetilde{\boldsymbol{x}}$ such that $\boldsymbol{A}\widetilde{\boldsymbol{x}} = \boldsymbol{b}$ and $\boldsymbol{f}(\widetilde{\boldsymbol{x}}) \le \epsilon + \boldsymbol{f}(\boldsymbol{x}^\star)$ in $\widetilde{O}\left(MRm^{1/3}\log(MR)\log\left(\frac{\boldsymbol{f}(\boldsymbol{x}^{(0)}) - \boldsymbol{f}(\boldsymbol{x}^\star)}{\epsilon}\right)\right)$ calls to a linear system solver.*

Resulting from the theorem above, as detailed in Section 5, are guarantees given by Theorems 5.4 and 5.5 which establish convergence rates of $\widetilde{O}(p^2\mu^{-1/(p-2)}m^{1/3}R)$ and $\widetilde{O}(m^{1/3}R)$, respectively, for $\mu$-regularized $\ell_p$ regression and logistic regression. We emphasize that the latter is, to our knowledge, the first such use of width reduction for directly solving constrained logistic regression problems.

## 1.2 Related Works

**Quasi-self-concordance and higher-order smoothness.** [Bac10] showed how to analyze Newton's method for quasi-self-concordant functions, with an emphasis on its application to logistic regression. Later, notions of local, or Hessian, stability which follow from quasi-self-concordance gave rise to methods with better dependence on various conditioning parameters along with a linear rate of convergence [KSJ18, MFBR19, CJJ$^+$20]. In the work by [KSJ18], the authors show how a trust-region-based Newton method [NW06] achieves linear convergence for locally stable functions without requiring, e.g., strong convexity. Meanwhile, after noting that quasi-self-concordance implies Hessian stability, [CJJ$^+$20] further improve the dependence on the distance to the optimum by leveraging Monteiro-Svaiter acceleration [MS13], which has proven useful in the context of near-optimal methods for higher-order acceleration [ASS19, GDG$^+$19]. However, in general these methods, which

assume higher-order smoothness, require access to an oracle which minimizes a higher-order Taylor expansion model, though in some cases this may be relaxed to requiring linear system solves [Bul20].

**Width reduction and $\ell_p$ regression.** The notion of width is common in the multiplicative weights literature [PST95, Fle00, GK07]. Most of these algorithms repeatedly solve a certain subproblem, and "width" is defined as an upper bound on the $\ell_\infty$-norm of the solution to these subproblems. The runtime of such algorithms depends linearly on the width, and since this quantity can have a large value, several approaches have been proposed to reduce the width.

The technique of width-reduction first came to prominence in seminal work by [CKM+11] for achieving faster approximate maximum flow, being the first to achieve an improved $m^{1/3}$ dependence. At a high level, the idea behind the approach is to solve a sequence of weighted $\ell_2$-minimizing flow subproblems, whereby at each iteration one of two cases occurs: either the proposed step is added to the current solution (a "flow" step) along with the weights, or else there exist some set of coordinates that exceed a certain threshold, and so their weights are updated accordingly (a "width reduction" step). Several works have since adapted this approach to regression problems [CMMP13, AKPS19, EV19, ABKS21] and matrix scaling [AZLOW17]. In particular, when comparing with [EV19], we note that the update steps for the weights are different from our algorithm. We also note that the number of width reduction steps in our algorithm are restricted by $\epsilon^{-2/3}$ iterations, which is similar to [EV19], but our algorithm requires $\epsilon^{-5/3}$ flow steps.

In addition to their importance in machine learning, regression methods capture several fundamental problems in scientific computing and signal processing. A recent line of work initiated by [BCLL18] showed how to attain high-accuracy solutions for $\ell_p$ regression using $O_p(m^{1/2-1/p})$ linear system solves, thus going beyond what is achievable via self-concordance. Building on this work, [AKPS19] showed how width reduction could be applied to this setting to achieve, as in the case of approximate maximum flow [CKM+11], a similar improvement from $O_p(m^{1/2})$ to $O_p(m^{1/3})$ (for $p \to \infty$). Further developments by [KPSW19, AS20] for graph problems showed almost-linear time solutions for $\ell_p$ regression for $p \approx \sqrt{\log(n)}$ which have since been a critical part of recent advances in high-accuracy maximum flow on unit-capacity graphs [LS20, KLS20].

**Accelerated methods.** Recent developments by [CJJ+20] have shown several advantages that arise in the case of unconstrained minimization for smooth, quasi-self-concordant problems. By considering a certain ball oracle method (whereby each call to the oracle returns the minimizer of the function inside an $\ell_2$ ball of radius $r$), [CJJ+20] implement an accelerated scheme which returns a solution to the unconstrained smooth convex minimization problem in $(R/r)^{2/3}$ calls to the oracle, where $R$ is the initial $\ell_2$-norm distance to the optimum, and they further show a matching lower bound under this oracle model. We note here that while our method obtains rates in terms of the $\ell_\infty$-norm of the optimum rather than the $\ell_2$-norm, in the worst case, we can have $\|x^\star\|_2 = \sqrt{m}\|x^\star\|_\infty$, which results in both rates being essentially the same.

While the approach of [CJJ+20] transfers its difficulty to implementing the oracle, a key insight from their work involves showing this can be done efficiently for smooth quasi-self-concordant functions when $r$ is sufficiently small, where the allowed size depends on the quasi-self-concordance parameter. One limitation to their results is that they apply directly to *unconstrained* optimization problems and require the function to be smooth, and so we complement these results in the quasi-self-concordant setting by establishing comparable rates for directly optimizing a large class of *constrained* convex problems without requiring smoothness of the function.

### 1.3 Outline of the Paper

After establishing the potential functions at the heart of our width reduction techniques, we present in Section 3 our oracle for roughly approximating a solution to problem (1). We then show in Section 4 how we may attain a high-accuracy solution by using the crude approximation as a starting point. Here, the key idea is to considering a sequence of optimization problems inside $\ell_\infty$ balls of manageable size, similar to [CMTV17, CJJ+20]. As in the case of the crude oracle, our primary advantage comes from carefully handling a pair of coupled potentials which are amenable to the large class of quasi-self-concordant problems, and in Section 5 we further show how our results may be applied to several problems of interest, including logistic and $\ell_p$ regression.

## 2 Preliminaries

**Notation:** We use boldface lowercase letters to denote vectors or functions and boldface uppercase letters for matrices. Scalars are non-bold letters. Our functions are univariate, and we overload function notation to act on a vector coordinate-wise, i.e. $\boldsymbol{f}(\boldsymbol{x}) = \sum_i \boldsymbol{f}(\boldsymbol{x}_i)$. The notation $\boldsymbol{x} \geq \boldsymbol{y}$ for vectors refers to entry-wise inequality. Refer to the algorithm boxes for definitions of certain algorithm specific parameters that appear in lemma and theorem statements.

### 2.1 Quasi-Self-Concordance

**Definition 2.1** (g.s.c. and q.s.c.). *Let $\boldsymbol{f} : \mathbb{R} \to \mathbb{R}$ be a thrice differentiable function with continuous third derivative, and let $\nu > 0$ and $M > 0$. We say that $\boldsymbol{f}$ is $(M, \nu)$-general-self-concordant (g.s.c.) if*

$$\forall x, \quad |\boldsymbol{f}'''(x)| \leq M\boldsymbol{f}''(x)^{\frac{\nu}{2}}.$$

*When $\nu = 2$, we have the following condition:*

$$\forall x, \quad |\boldsymbol{f}'''(x)| \leq M\boldsymbol{f}''(x),$$

*and we call such functions $M$-quasi-self-concordant (q.s.c.).*

### 2.2 Problem

Recall that we are solving the following problem:

$$\min_{\boldsymbol{A}\boldsymbol{x}=\boldsymbol{b}} \sum_i \boldsymbol{f}\big((\boldsymbol{P}\boldsymbol{x})_i\big),$$

where $\boldsymbol{A} \in \mathbb{R}^{d \times n}, \boldsymbol{b} \in \mathbb{R}^d, \boldsymbol{P} \in \mathbb{R}^{m \times n}, d \leq n$, and $m \geq n$, and such that $\boldsymbol{f}$ is convex, $M$-q.s.c. and, for $\boldsymbol{w} \geq \boldsymbol{w}_0 \geq 0$, $\boldsymbol{f}''(\boldsymbol{w}_i)$ is monotonic $\forall i$. We can ignore the case when $\boldsymbol{f}''$ is constant since that corresponds to a quadratic problem which we know how to solve directly via linear system solves.

**Assumptions on the Optimum $\boldsymbol{x}^\star$**

We assume that $R \in \mathbb{R}_{>0}$ is such that the optimum $\boldsymbol{x}^\star \overset{\text{def}}{=} \arg\min_{\boldsymbol{A}\boldsymbol{x}=\boldsymbol{b}} \boldsymbol{f}(\boldsymbol{P}\boldsymbol{x})$ satisfies

$$\|\boldsymbol{P}\boldsymbol{x}^\star\|_\infty \leq R. \tag{2}$$

We now define the potentials that we track in the algorithm.

### 2.3 Potentials

**Definition 2.2** (Dual Potential). *For a weights vector $\boldsymbol{w} \in \mathbb{R}_{\geq 0}^m$, we define a potential*

$$\Phi(\boldsymbol{w}) = \sum_i \Phi(\boldsymbol{w}_i) = \sum_i \boldsymbol{f}''(\boldsymbol{w}_i).$$

We also define the following corresponding potential, which gives rise to the linear regression problem that we will need to solve at each step of our algorithm.

**Definition 2.3** (Resistances and Corresponding Potential). *For a weights vector $\boldsymbol{w} \in \mathbb{R}_{\geq 0}^m$ and $\epsilon > 0$, define resistances $\boldsymbol{r} \in \mathbb{R}_{\geq 0}^m$ and a corresponding potential $\Psi$ as,*

$$\boldsymbol{r}_i = \frac{1}{R^2}\left(\boldsymbol{f}''(\boldsymbol{w}_i) + \frac{\epsilon\Phi(\boldsymbol{w})}{m}\right),$$

$$\Psi(\boldsymbol{r}) = \min_{\boldsymbol{A}\Delta=\boldsymbol{b}} \sum_i \boldsymbol{r}_i(\boldsymbol{P}\Delta)_i^2.$$

We have the following relation between our two potentials $\Phi$ and $\Psi$.

**Lemma 2.4.** *For $\epsilon > 0$, resistances $\boldsymbol{r}$ (Definition 2.3), with corresponding weights $\boldsymbol{w}$, we have*

$$\Psi(\boldsymbol{r}) \leq (1+\epsilon)\Phi(\boldsymbol{w}).$$

*In addition, letting $\|\boldsymbol{P}\|_{\min} = \min_{\boldsymbol{A}\boldsymbol{x}=\boldsymbol{b}} \|\boldsymbol{P}\boldsymbol{x}\|_2$ and $\|\boldsymbol{A}\|$ denote the operator norm of $\boldsymbol{A}$, we have*

$$\Psi(\boldsymbol{r}) \geq \frac{\epsilon\Phi(\boldsymbol{w})}{mR^2} \frac{\|\boldsymbol{P}\|_{\min}^2\|\boldsymbol{b}\|_2^2}{\|\boldsymbol{A}\|^2} \overset{\text{def}}{=} \Phi(\boldsymbol{w})L.$$

# 3 Algorithm and Analysis for a Crude Solution for Q.S.C. Functions

In this section, we give an algorithm for solving Problem (1) to a crude approximation; namely, we return a solution $\widetilde{x}$ such that $A\widetilde{x} = b$, i.e., it satisfies the subspace constraints, and $\|P\widetilde{x}\|_\infty$ is bounded. We will later see in our applications how this translates into a constant or polynomial approximation guarantee to the function value for some functions. In the next section we will see how we can use the guarantees of the solution returned as a starting solution and boost it to an $\epsilon$ approximate solution.

Our algorithm is based on combining a multiplicative weight update (MWU) scheme with width reduction. Though such algorithms have so far only been used for $\ell_p$-regression, $p = 1$ or $p \in [2, \infty]$, here we are able to extend the analysis to q.s.c. functions, while also providing a unified analysis for the known cases of $\ell_p$-regression (refer to Section 5 to see how we apply this algorithm to these instances). We note that we can extend this analysis to other general-self-concordant functions, and we have deferred these cases to the appendix.

## 3.1 Algorithm and Analysis

We describe our width-reduced multiplicative weight update method in Algorithm 1. We note that the width of $|P\Delta|$ is being reduced, though the weight updates are not entirely multiplicative. For a width step it is multiplicative in $f''(w)$ (lines 14-18), but for a flow step (line 11) we perform a purely additive update directly on the weights.

Our proof relies on tracking two potentials, $\Psi$ (Definition 2.3) and $\Phi$ (Definition 2.2) that depend on the weights. We first show how these potentials change with weight updates corresponding to a flow step and a width reduction step in the algorithm. We next show that if our algorithm runs for at most $K = \widetilde{O}(m^{1/3})$ width reduction steps, then after $T = \widetilde{O}(m^{1/3})$ flow steps we can bound $\Phi$. Further, using the relation between $\Phi$ and $\Psi$ (Lemma 2.4) and appropriately chosen parameters, we show that we cannot have more than $K$ width reduction steps. The key part of the analysis lies in the growth of $\Phi$ with respect to both flow and width steps.

---

**Algorithm 1** Width-Reduced Algorithm for $M$-q.s.c. Functions

---

1: **procedure** QSC-MWU($A, b, P, M, R, \epsilon$)
2:     $x^{(0,0)} = 0, w^{(0,0)} = w_0$ ($\Phi'(w)$ monotonic for $w \geq w_0$, $\Phi(w_0) > 0$)
3:     $\tau \leftarrow \widetilde{\Theta}\left(m^{1/3}\epsilon^{-2/3}\right)$
4:     $\alpha \leftarrow \widetilde{\Theta}\left(m^{-1/3}M^{-1}\epsilon^{2/3}\right)$
5:     $t = 0, k = 0, \quad T = \alpha^{-1}M^{-1}\epsilon^{-1} = \widetilde{\Theta}(m^{1/3}\epsilon^{-5/3})$
6:     **while** $t \leq T$ **do**
7:         $r_i^{(t,k)} \leftarrow \frac{1}{R^2}\left(f''(w_i^{(t,k)}) + \frac{\epsilon\Phi(w^{(t,k)})}{m}\right)$                                 ▷ Resistances
8:         $\widetilde{\Delta} \leftarrow \arg\min_{A\Delta=b} \sum_i r_i(P\Delta)_i^2$                                         ▷ Oracle
9:         **if** $\left\|P\widetilde{\Delta}\right\|_\infty \leq R\tau$ **then**                                           ▷ Flow Step
10:             $x^{(t+1,k)} \leftarrow x^{(t,k)} + \widetilde{\Delta}$
11:             $w^{(t+1,k)} \leftarrow w^{(t,k)} + \frac{\epsilon\alpha}{R}|P\widetilde{\Delta}|$
12:             $t \leftarrow t + 1$
13:         **else**
14:             **for** Indices $i$ such that $|P\widetilde{\Delta}|_i \geq R\tau$ **do**                   ▷ Width Reduction
15:                 **if** $f''$ is non-decreasing in $w$ **then**[1]
16:                     $w^{(t,k+1)}$ is such that $r_i^{(t,k+1)} \leftarrow (1+\epsilon)r_i^{(t,k)}$
17:                 **else**
18:                     $w^{(t,k+1)}$ is such that $r_i^{(t,k+1)} \leftarrow \frac{1}{1+\epsilon}r_i^{(t,k)}$
19:             $k \leftarrow k + 1$
20:     **return** $x^{(T,k)}/T$

---

[1] We will see later how such weight/resistance changes can be realized for some special cases.

**Changes in $\Psi$ and $\Phi$**

Here we show how the potentials $\Phi$ and $\Psi$ change with flow and width reduction steps, and we defer the proofs to the appendix.

**Lemma 3.1.** *Let $\Psi$ be as defined in 2.3. After $t$ flow steps and $k$ width reduction steps, we have,*

$$\Psi(\boldsymbol{r}^{(t,k)}) \geq \Psi(\boldsymbol{r}^{(0,0)}) \left(1 + \frac{\epsilon^2 \tau^2}{(1+\epsilon)^2 m}\right)^k \qquad \text{if } \boldsymbol{f}'' \text{ non-decreasing in } \boldsymbol{w},$$

$$\Psi(\boldsymbol{r}^{(t,k)}) \leq \Psi(\boldsymbol{r}^{(0,0)}) \left(1 - \frac{\epsilon^2 \tau^2}{2(1+\epsilon)^2 m}\right)^k \qquad \text{if } \boldsymbol{f}'' \text{ non-increasing in } \boldsymbol{w}.$$

**Lemma 3.2.** *Suppose $\boldsymbol{f}$ is $M$-q.s.c. Let $\alpha$ and $\tau$ be such that $\alpha\tau \leq M^{-1}$. After $t$ flow steps and $k$ width reduction steps, our potential $\Phi$ satisfies*

$$\Phi(\boldsymbol{w}^{(t,k)}) \leq \left(1 + \epsilon(1+\epsilon)^2 \alpha M\right)^t \left(1 + \epsilon(1+\epsilon)\tau^{-1}\right)^k \Phi(\boldsymbol{w}_0) \quad \text{if } \boldsymbol{f}'' \text{ non-decreasing in } \boldsymbol{w},$$

$$\Phi(\boldsymbol{w}^{(t,k)}) \geq \left(1 - \epsilon(1+\epsilon)^2 \alpha M\right)^t \left(1 - \epsilon(1+\epsilon)\tau^{-1}\right)^k \Phi(\boldsymbol{w}_0) \quad \text{if } \boldsymbol{f}'' \text{ non-increasing in } \boldsymbol{w}.$$

**Runtime Bound**

We now establish the final rate of convergence for Algorithm 1.

**Theorem 3.3.** *Let $\epsilon > 0$, $\boldsymbol{f}$ be $M$-q.s.c. After $T \leq \frac{\alpha^{-1}}{M\epsilon} = \widetilde{\Theta}(m^{1/3}\epsilon^{-5/3})$ flow steps and $K \leq \tau = \widetilde{\Theta}(m^{1/3}\epsilon^{-2/3})$ width reduction steps, Algorithm 1 returns $\widetilde{\boldsymbol{x}}$ such that $\boldsymbol{A}\widetilde{\boldsymbol{x}} = \boldsymbol{b}$, $\|\boldsymbol{P}\widetilde{\boldsymbol{x}}\|_\infty \leq RM\|\boldsymbol{w}^{(T,K)}\|_\infty$, where $\boldsymbol{w}^{(T,K)}$ is the final weights vector that satisfies:*

$$\Phi(\boldsymbol{w}^{(T,K)}) \leq \Phi(\boldsymbol{w}_0)e^{1+4\epsilon} \qquad \text{if } \boldsymbol{f}'' \text{ is non-decreasing in } \boldsymbol{w},$$

$$\Phi(\boldsymbol{w}^{(T,K)}) \geq \Phi(\boldsymbol{w}_0)e^{-(1+4\epsilon)} \qquad \text{if } \boldsymbol{f}'' \text{ is non-increasing in } \boldsymbol{w}.$$

*Proof.* We show the case when $\boldsymbol{f}''$ is a non-decreasing function. The other case follows similarly. We set,

$$\tau \leftarrow \widetilde{\Theta}\left(m^{1/3}\epsilon^{-2/3}\right) \quad \alpha \leftarrow \widetilde{\Theta}\left(m^{-1/3}M^{-1}\epsilon^{2/3}\right).$$

After $T = \frac{\alpha^{-1}}{M\epsilon}$ flow steps and $K = \tau$ width reduction steps, from Lemma 3.2, we have,

$$\Phi(\boldsymbol{w}^{(T,K)}) \leq \left(1 + \epsilon(1+\epsilon)^2 \alpha M\right)^T \left(1 + \epsilon(1+\epsilon)\tau^{-1}\right)^K \Phi(\boldsymbol{w}_0)$$

$$\leq \Phi(\boldsymbol{w}_0)e^{\epsilon(1+\epsilon)^2\alpha MT + \epsilon(1+\epsilon)\tau^{-1}K} \leq \Phi(\boldsymbol{w}_0)e^{(1+4\epsilon)}.$$

We now show we cannot have more width steps. Throughout the algorithm, we have $\Phi(\boldsymbol{w}^{(t,k)}) \leq \Phi(\boldsymbol{w}_0)e^{1+4\epsilon}$. From Lemma 2.4 we always have $\Psi(\boldsymbol{r}^{(0,0)}) \geq \Phi(\boldsymbol{w}_0)L$ and $\Psi(\boldsymbol{r}^{(T,K)}) \leq (1+\epsilon)\Phi(\boldsymbol{w}^{(T,K)}) \leq (1+\epsilon)e^{1+4\epsilon}\Phi(\boldsymbol{w}_0)$. Thus, from Lemma 3.1, we must have,

$$(1+\epsilon)e^{1+4\epsilon}\Phi(\boldsymbol{w}_0) \geq L\Phi(\boldsymbol{w}_0)\left(1 + \frac{\epsilon^2\tau^2}{(1+\epsilon)^2 m}\right)^K,$$

From the definition of $\tau$, we note that $K$ has to be less than $\tau$ for the above bound to be satisfied. Next, let $\widetilde{\Delta}^{(t)}$ denote the solution of our oracle at iteration $t$ of the flow step. From the $\boldsymbol{x}$ and $\boldsymbol{w}$ update in the algorithm,

$$|\boldsymbol{P}\widetilde{\boldsymbol{x}}| = \left|\sum_t \boldsymbol{P}\widetilde{\Delta}^{(t)}\right|\epsilon\alpha M \leq \frac{\epsilon\alpha}{R}\sum_t \left|\boldsymbol{P}\widetilde{\Delta}^{(t)}\right|RM \leq \boldsymbol{w}^{(T,K)}RM.$$

This concludes our proof. $\qquad\square$

# 4 Boosting to a High-Accuracy Solution for Q.S.C. Functions

In this section, we give a width-reduced multiplicative weights update algorithm that, given a starting solution $\boldsymbol{x}^{(0)}$ satisfying $\|\boldsymbol{x}^{(0)}\|_\infty \leq R$ and $\boldsymbol{A}\boldsymbol{x}^{(0)} = \boldsymbol{b}$, finds $\widetilde{\boldsymbol{x}}$ such that $\boldsymbol{A}\widetilde{\boldsymbol{x}} = \boldsymbol{b}$ and $\boldsymbol{f}(\widetilde{\boldsymbol{x}}) \leq (1+\epsilon)\boldsymbol{f}(\boldsymbol{x}^\star)$ for any q.s.c. function $\boldsymbol{f}$. We would mention here that for the algorithms in this section, it is key that we have a starting solution that satisfies our subspace constraints and has $\ell_\infty$-norm bounded by $R$. Thus, the algorithms here may be of independent interest if such a starting solution is available. We can otherwise use Algorithm 1 with $\epsilon = 1$ to obtain such a solution.

For any $\boldsymbol{x}$, we define a residual problem, and we show how it is sufficient to solve the residual problem approximately $\log(\epsilon^{-1})$ times to obtain our high-accuracy solution. Similar approaches have been applied to specific functions such as softmax [AZLOW17] and $\ell_p$-regression [AKPS19]. We unify these approaches and give a version that works for any q.s.c. function.

We further note that, in the spirit of [AZLOW17], our residual problem is to optimize a simple quadratic objective inside an $\ell_\infty$ box. The difficulty lies in solving such $\ell_\infty$ box constraints fast. We use a binary search followed by a width-reduced multiplicative weights routine analogous to [CKM$^+$11] to solve our residual problem.

**Definition 4.1** (Residual Problem). *We define the residual objective at any $\boldsymbol{x}$ satisfying $\|\boldsymbol{P}\boldsymbol{x}\|_\infty \leq R$ as*

$$res(\Delta) = \nabla \boldsymbol{f}(\boldsymbol{x})^\top \boldsymbol{P}\Delta - e^{-1}(\boldsymbol{P}\Delta)^\top \nabla^2 \boldsymbol{f}(\boldsymbol{x})\boldsymbol{P}\Delta,$$

*and the residual problem as*

$$\max_\Delta \quad res(\Delta)$$
$$s.t. \quad \boldsymbol{A}\Delta = 0, \quad and \quad \|\boldsymbol{P}\Delta - \boldsymbol{z}\|_\infty \leq \frac{1}{2M}. \tag{3}$$

*Here, $\boldsymbol{z}$ is a vector that depends on $\boldsymbol{x}$, and is defined as*

$$\boldsymbol{z}_i = \begin{cases} \left(-\frac{1}{2M} + R + (\boldsymbol{P}\boldsymbol{x})_i\right) \in [-\frac{1}{2M}, 0)], & \text{if } (\boldsymbol{P}\boldsymbol{x})_i - \frac{1}{2M} < -R \\ \left(-R + (\boldsymbol{P}\boldsymbol{x})_i + \frac{1}{2M}\right) \in (0, \frac{1}{2M}], & \text{if } (\boldsymbol{P}\boldsymbol{x})_i + \frac{1}{2M} > R \\ 0, & \text{otherwise.} \end{cases}$$

We note that any solution $\Delta$ satisfying the above box constraint satisfies $\|\boldsymbol{P}\Delta\|_\infty \leq M^{-1}$ and $\left\|\boldsymbol{P}\boldsymbol{x} - e^{-2}\boldsymbol{P}\Delta\right\|_\infty \leq R$.

**Lemma 4.2.** *[Iterative Refinement] Let $\boldsymbol{f}$ be $M$-q.s.c. and $\widetilde{\Delta}^{(t)}$ a $\kappa$-approximate solution to the residual problem at $\boldsymbol{x}^{(t)}$ (Problem (3)). Starting from $\boldsymbol{x}^{(0)}$ such that $\boldsymbol{A}\boldsymbol{x}^{(0)} = \boldsymbol{b}$, $\|\boldsymbol{x}^{(0)}\|_\infty \leq R$, and iterating as $\boldsymbol{x}^{(t+1)} = \boldsymbol{x}^{(t)} - e^{-2}\widetilde{\Delta}^{(t)}$, after at most $O\left(\kappa MR \log\left(\frac{\boldsymbol{f}(\boldsymbol{x}^{(0)}) - \boldsymbol{f}(\boldsymbol{x}^\star)}{\epsilon}\right)\right)$ iterations we get $\boldsymbol{x}$ such that $\boldsymbol{A}\boldsymbol{x} = \boldsymbol{b}$ and $\boldsymbol{f}(\boldsymbol{x}) \leq \boldsymbol{f}(\boldsymbol{x}^\star) + \epsilon$.*

## 4.1 Approximately Solving the Residual Problem

### Binary Search

**Lemma 4.3.** *Let $\nu$ be such that $\boldsymbol{f}(\boldsymbol{x}^{(t)}) - \boldsymbol{f}(\boldsymbol{x}^\star) \in (\nu/2, \nu]$ and $\Delta^\star$ denote the optimum of the residual problem at $\boldsymbol{x}^{(t)}$. Then, $res(\Delta^\star) \in \left(\frac{\nu}{8MR}, e^2\nu\right]$.*

From the above lemma we may do a binary search in the range $\left(\frac{\nu}{8MR}, e^2\nu\right]$. Let us start with the assumption that the residual problem has a solution between $(\zeta/2, \zeta]$.

**Lemma 4.4.** *Let $\zeta$ be such that $res(\Delta^\star) \in (\zeta/2, \zeta]$ and $\Delta^\star$ the optimum of the residual problem. Then, $(\boldsymbol{P}\Delta^\star)^\top \nabla^2 \boldsymbol{f}(\boldsymbol{x})\boldsymbol{P}\Delta^\star \leq e \cdot \zeta$.*

### Using Width Reduction

We will show that Algorithm 3 returns $\Delta$ such that $\|\boldsymbol{P}\Delta - \boldsymbol{z}\|_\infty \leq \frac{1}{2M}$ and $res(\Delta) \geq \frac{1}{400}\zeta$.

---

**Algorithm 2** Boosting to $\epsilon$-approximation

---

1: **procedure** QSC-MIN($(\boldsymbol{A}, \boldsymbol{b}, \boldsymbol{P}, \boldsymbol{x}_0, M, \epsilon)$ such that $\boldsymbol{A}\boldsymbol{x}_0 = \boldsymbol{b}$, $\|\boldsymbol{x}_0 - \boldsymbol{x}^\star\|_\infty \leq 2R$)
2:     $\boldsymbol{x}^{(0)} = \boldsymbol{x}_0, \tau \leftarrow m^{1/3}, \alpha \leftarrow m^{-1/3}$
3:     **for** $i \leq O(MR \log \epsilon^{-1})$ **do**
4:         **for** $\nu \in \big(\epsilon, \boldsymbol{f}(\boldsymbol{x})\big]$ **do**                    ▷ Decrease $\nu$ by 2 in each iteration
5:             **for** $\zeta \in \big(\frac{\nu}{8MR}, e^2\nu\big]$ **do**                    ▷ Decrease $\zeta$ by 2 in each iteration
6:                 $\boldsymbol{y}_{\zeta,\nu} \leftarrow MWU(\boldsymbol{A}, \boldsymbol{P}, \boldsymbol{x}^{(i)}, M, \zeta)$
7:         $\boldsymbol{x}^{(i+1)} \leftarrow \boldsymbol{x}^{(i)} - e^{-2} \arg\min_{\boldsymbol{y}_{\zeta,\nu}} \boldsymbol{f}(\boldsymbol{x} - e^{-2}\boldsymbol{y}_{\zeta,\nu})$

---

**Algorithm 3**

---

1: **procedure** MWU($\boldsymbol{A}, \boldsymbol{P}, \boldsymbol{x}, M, \zeta$)
2:     $\boldsymbol{y}^{(0)} = 0, \boldsymbol{w}^{(0)} = \frac{\zeta}{m}$
3:     $t = 0$
4:     $\boldsymbol{A}' = \left[\boldsymbol{A}^\top, \boldsymbol{P}^\top \nabla \boldsymbol{f}(\boldsymbol{x})\right]^\top, \boldsymbol{b} = [0, \frac{\zeta}{2}]$
5:     **while** $\|\boldsymbol{w}\|_1 \leq 10\zeta$ **do**
6:         $\widetilde{\Delta} \leftarrow \arg\min_{\boldsymbol{A}'\Delta=b'} \sum_j \boldsymbol{f}''(\boldsymbol{x}_j)(\boldsymbol{P}\Delta)_j^2 + 4M^2 \sum_j \left(\boldsymbol{w}_j^{(t)} + \frac{\|\boldsymbol{w}^{(t)}\|_1}{m}\right)(\boldsymbol{P}\Delta - \boldsymbol{z})_j^2$
7:         **if** $2M\left\|\boldsymbol{P}\widetilde{\Delta} - \boldsymbol{z}\right\|_\infty \leq \tau$ **then**                    ▷ Flow Step
8:             $\boldsymbol{y}^{(t+1)} \leftarrow \boldsymbol{y}^{(t)} + \widetilde{\Delta}$
9:             $\boldsymbol{w}^{(t+1)} \leftarrow \boldsymbol{w}^{(t)}\left(1 + \frac{1}{2}\alpha M|\boldsymbol{P}\widetilde{\Delta} - \boldsymbol{z}|\right)$
10:        **else**
11:            **for** Indices $i$ such that $2M|\boldsymbol{P}\widetilde{\Delta} - \boldsymbol{z}|_i \geq \tau$ **do**
12:                $\boldsymbol{w}_i^{(t+1)} \leftarrow 2\boldsymbol{w}_i^{(t)}$                    ▷ Width Step
13:        $t \leftarrow t + 1$
14:    **return** $\frac{\boldsymbol{y}^{(t)}}{100t}$

---

**Lemma 4.5.** *Let $\zeta$ be such that $res(\Delta^\star) \in (\zeta/2, \zeta]$. Algorithm 3 returns $\boldsymbol{y}$ such that $\boldsymbol{A}\boldsymbol{y} = 0$, $\|\boldsymbol{P}\boldsymbol{y} - \boldsymbol{z}\|_\infty \leq \frac{1}{2M}$ and $res(\boldsymbol{y}) \geq \frac{1}{400}res(\Delta^\star)$ in $O(m^{1/3})$ calls to a linear system solver.*

We now state the main result of the section which follows directly from Lemmas 4.2, 4.3 and 4.5

**Theorem 4.6.** *For $\epsilon > 0$, $M$-q.s.c. function $\boldsymbol{f}$ and, $\boldsymbol{x}^{(0)}$ such that $\boldsymbol{A}\boldsymbol{x}^{(0)} = \boldsymbol{b}$, $\|\boldsymbol{x}^{(0)}\|_\infty \leq R$, Algorithm 2 finds $\widetilde{\boldsymbol{x}}$ such that $\boldsymbol{A}\widetilde{\boldsymbol{x}} = \boldsymbol{b}$ and $\boldsymbol{f}(\widetilde{\boldsymbol{x}}) - \boldsymbol{f}(\boldsymbol{x}^\star) \leq \epsilon$ in $\widetilde{O}\left(MRm^{1/3}\log(MR)\log\left(\frac{\boldsymbol{f}(\boldsymbol{x}^{(0)}) - \boldsymbol{f}(\boldsymbol{x}^\star)}{\epsilon}\right)\right)$ calls to a linear system solver.*

## 5   Applications

We now show how our methods may be applied to various quasi-self-concordant functions.

### 5.1   Sum of Exponentials, Softmax and $\ell_\infty$-regression

We recall the softmax function $\mathrm{smax}_\nu(\boldsymbol{P}\boldsymbol{x}) = \nu \log\left(\sum_i e^{\frac{(\boldsymbol{P}\boldsymbol{x})_i}{\nu}}\right)$, which we may note is $1/\nu$-q.s.c. We start by assuming that at the optimum, for $R \geq \Omega\big((\log m)^{-1}\big)$, $\mathrm{smax}_\nu(\boldsymbol{P}\boldsymbol{x}^\star) \leq R$.

We apply Algorithm 1 to $\sum_i e^{\frac{(\boldsymbol{P}\boldsymbol{x})_i}{\nu}}$, which is also $1/\nu$-q.s.c. We can use the following weight update step for the width reduction step: $\boldsymbol{w}_i^{(t,k+1)} \leftarrow \boldsymbol{w}_i^{(t,k)} + \nu \log(1 + \epsilon)$.

**Theorem 5.1.** *Let $x^\star$ denote the optimum of $\min_{Ax=b} \mathrm{smax}_\nu(Px)$. Algorithm 1, when applied to the function $f(Px) = \sum_i e^{\frac{(Px)_i}{\nu}}$, returns $\widetilde{x}$ such that $A\widetilde{x} = b$, and*

$$\mathrm{smax}_t(P\widetilde{x}) \leq (1 + \widetilde{O}(\nu))\mathrm{smax}_\nu(Px^\star),$$

*in at most $\widetilde{O}(m^{1/3}\nu^{-5/3})$ calls to a linear system solver.*

*Proof.* We know that $\mathrm{smax}_\nu(P\widetilde{x}) \leq \|P\widetilde{x}\|_\infty + \nu\log m$. From Lemma 3.3, we have that $\widetilde{x}$ is obtained in at most $\widetilde{O}(m^{1/3}\nu^{-5/3})$ calls to a linear system solver satisfying $A\widetilde{x} = b$. Further, we also have, $\|P\widetilde{x}\|_\infty \leq MR\|w^{(T,K)}\|_\infty = R\frac{\|w^{(T,K)}\|_\infty}{\nu}$. We will now bound $\frac{\|w^{(T,K)}\|_\infty}{\nu}$. We note that $\Phi(w^{(T,K)}) \leq \Phi(w_0)e^{1+4\nu}$. For $w_0 = 0$,

$$\Phi(w^{(T,K)}) = \frac{1}{\nu^2}\sum_i e^{\frac{w_i^{(T,K)}}{\nu}} = \Phi(w_0)\sum_i e^{\frac{w_i^{(T,K)}}{\nu}} \leq \Phi(w_0)e^{1+4\nu}.$$

Therefore, we must have $w^{(T,K)} \leq (1+4\nu)\nu$. Our bound is

$$\mathrm{smax}_\nu(P\widetilde{x}) \leq (1+4\nu)R + \nu\log m \leq \left(1 + \widetilde{O}(\nu)\right)R,$$

for $R \geq \Omega(1/\log m)$. We can now do a binary search on $R$ as follows to obtain

$$\mathrm{smax}_\nu(P\widetilde{x}) \leq (1 + \widetilde{O}(\nu))\mathrm{smax}_\nu(Px^\star).$$

**Binary search on $R$:** Let $R_0$ denote the value $\|Px^\star\|_\infty = R_0$. Now, for any $R \geq R_0$, we attain an $\widetilde{x}$ which has an objective value at most $R(1 + 4\nu)$. For any $R < R_0$, as long as $R$ is such that the plane $Ax = b$ has at least one point with infinity norm at most $R$, we will get a feasible solution to our problem. However, the objective value guarantee of $R(1 + 4\nu)$ may not hold. Since the optimum is $R_0$, the solution returned in such cases must give an objective value larger than $R_0$. We can thus do a binary search on $R$ and reach $O(\nu)$ close to the value $R_0$. This will require running our algorithm $O\left(\log(R_0\nu^{-1})\right)$ times. In the end we can return the $x$ which gives the smallest objective values among all these runs. $\qquad\square$

**Theorem 5.2.** *Let $x^\star$ denote the optimum of the $\ell_\infty$-regression problem, $\min_{Ax=b} \|Px\|_\infty$. Algorithm 1 when applied to the function $f(Px) = \sum_i \left(e^{\frac{(Px)_i}{\nu}} + e^{\frac{-(Px)_i}{\nu}}\right)$ for $\nu = \Omega\left(\frac{\epsilon}{\log m}\right)$, returns $\widetilde{x}$ such that $A\widetilde{x} = b$ and*

$$\|P\widetilde{x}\|_\infty \leq (1 + \epsilon)\|Px^\star\|_\infty,$$

*in at most $\widetilde{O}(m^{1/3}\epsilon^{-5/3})$ calls to a linear system solve.*

**Theorem 5.3.** *For $\delta > 0$, let $\overline{x}$ be the solution returned by Algorithm 1 (with $\epsilon = 1$) applied to $f(Px) = \sum_i e^{\frac{(Px)_i}{\nu}}$. Now, Algorithm 2 with starting solution $x^{(0)} = \overline{x}$, applied to $f$ finds $\widetilde{x}$ such that $A\widetilde{x} = b$ and $\sum_i e^{\frac{(P\widetilde{x})_i}{\nu}} \leq (1 + \delta)\sum_i e^{\frac{(Px^\star)_i}{\nu}}$ in at most $O\left(m^{1/3}R^2\nu^{-2}\log\left(\frac{m}{\delta}\right)\right)$ calls to a linear system solver.*

## 5.2 $p$-Norm Regression

We will solve, $\min_{Ax=b} f(Px) = \|Px\|_p^p + \mu\|Px\|_2^2$, for $p \geq 3$ which is $p\mu^{-1/(p-2)}$-q.s.c. w.r.t. its argument. We first apply Algorithm 1 to this function and use the returned solution as the starting point of Algorithm 2. We can use the following weight update step for the width reduction step: $w_i^{(t,k+1)} \leftarrow (1+\epsilon)^{1/(p-2)}w_i^{(t,k)}$.

**Theorem 5.4.** *For $\delta > 0$ and $p \geq 3$, let $\overline{x}$ be the solution returned by Algorithm 1 (with $\epsilon = 1$) applied to $f(Px) = \|Px\|_p^p + \mu\|Px\|_2^2$. Now, Algorithm 2 with starting solution $x^{(0)} = \overline{x}$, applied to $f$ finds $\widetilde{x}$ such that $A\widetilde{x} = b$ and $f(P\widetilde{x}) \leq f(Px^\star) + \delta$ in at most $O\left(p^2\mu^{-1/(p-2)}m^{1/3}R\log\left(\frac{pmR}{\mu\delta}\right)\right)$ calls to a linear system solver.*

### 5.3 Logistic Regression

We consider the function $\boldsymbol{f}(\boldsymbol{Px}) = \sum_i \log(1 + e^{(\boldsymbol{Px})_i})$ which is 1-q.s.c. w.r.t its argument. We will use Algorithm 1 with the following weight update for the width reduction step which reduces the resistance by a factor of $(1 + \epsilon)$: $\boldsymbol{w}_i^{(t,k+1)} \leftarrow \boldsymbol{w}_i^{(t,k)} + 0.9\epsilon$

**Theorem 5.5.** *For $\delta > 0$, let $\overline{\boldsymbol{x}}$ be the solution returned by Algorithm 1 (with $\epsilon = 1$) applied to $\boldsymbol{f}(\boldsymbol{Px}) = \sum_i \log(1 + e^{(\boldsymbol{Px})_i})$. Now, Algorithm 2 with starting solution $\boldsymbol{x}^{(0)} = \overline{\boldsymbol{x}}$, applied to $\boldsymbol{f}$ finds $\widetilde{\boldsymbol{x}}$ such that $\boldsymbol{A}\widetilde{\boldsymbol{x}} = \boldsymbol{b}$ and $\sum_i \log(1 + e^{(\boldsymbol{P}\widetilde{\boldsymbol{x}})_i}) \leq \sum_i \log(1 + e^{(\boldsymbol{Px}^\star)_i}) + \delta$ in at most $O\left(m^{1/3}R\log\left(\frac{mR}{\delta}\right)\right)$ calls to a linear system solver.*

## Acknowledgments and Disclosure of Funding

SS is supported by a Discovery Grant awarded by NSERC (Natural Sciences and Engineering Research Council of Canada). DA is supported by a Post Graduate Doctoral Scholarship awarded by NSERC and SS's Discovery Grant.

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
