# A  Algorithm for General-Self-Concordant Functions

In this section we will show how to use our algorithms for the following classes of general-self-concordant functions.

1. $6 > \nu \geq 2$: $\boldsymbol{f}$ is $(N, \nu)$-g.s.c. and $L$-smooth.
2. $\nu < 2$: $\boldsymbol{f}$ is $(N, \nu)$-g.s.c., $L$-smooth and $\mu$-strongly convex.

We will use the following result to reduce these problems to $(M, 2)$-g.s.c. problems and use our algorithms.

**Lemma A.1** (Prop 4. [STD19]). *Let $\boldsymbol{f}$ be $(M, \nu)$-g.s.c. with $\nu > 0$. Then:*

(a) *If $\nu \in (0, 3]$ and $\boldsymbol{f}$ is also strongly convex with strong convexity parameter $\mu > 0$ in $\ell_2$-norm, then $\boldsymbol{f}$ is also $\left( \frac{M}{\sqrt{\mu}^{3-\nu}}, 3 \right)$-g.s.c.*

(b) *If $\nu \geq 2$ and $\nabla \boldsymbol{f}$ is Lipschitz continuous with finite Lipschitz constant $L$ in $\ell_2$-norm, then $\boldsymbol{f}$ is also $\left( ML^{\frac{\nu}{2} - 1}, 2 \right)$-g.s.c.*

We thus have the following result.

**Theorem A.2.** *For $\delta > 0$, $\boldsymbol{f}$ $(N, \nu)$-g.s.c. $6 > \nu \geq 2$ and $L$-smooth, let $\overline{\boldsymbol{x}}$ be the solution returned by Algorithm 1 (with $\epsilon = 1$) applied to $\boldsymbol{f}(\boldsymbol{x})$. Now, Algorithm 2 with starting solution $\boldsymbol{x}^{(0)} = \overline{\boldsymbol{x}}$, applied to $\boldsymbol{f}$ finds $\widetilde{\boldsymbol{x}}$ such that $\boldsymbol{A}\widetilde{\boldsymbol{x}} = \boldsymbol{b}$ and $\sum_i \boldsymbol{f}(\boldsymbol{P}\widetilde{\boldsymbol{x}}_i) \leq \sum_i \boldsymbol{f}(\boldsymbol{P}\boldsymbol{x}_i^\star) + \delta$ in at most*

$$
O\left( m^{1/3} N L^{\frac{\nu - 2}{2}} R \log\left( \frac{\boldsymbol{f}(\boldsymbol{x}^{(0)}) - \boldsymbol{f}(\boldsymbol{x}^\star))}{\delta} \right) \right)
$$

*calls to a linear system solver.*

*Proof.* From Lemma A.1, $\boldsymbol{f}$ is $(NL^{(\nu-2)/2}, 2)$-g.s.c. We now use Lemma 3.3 with $M = NL^{(\nu-2)/2}$ followed by Theorem 4.6. $\qquad \square$

**Theorem A.3.** *For $\delta > 0$, $\boldsymbol{f}$ $(N, \nu)$-g.s.c. $2 > \nu \geq 0$ and $L$-smooth $\mu$-strongly convex, let $\overline{\boldsymbol{x}}$ be the solution returned by Algorithm 1 (with $\epsilon = 1$) applied to $\boldsymbol{f}(\boldsymbol{x})$. Now, Algorithm 2 with starting solution $\boldsymbol{x}^{(0)} = \overline{\boldsymbol{x}}$, applied to $\boldsymbol{f}$ finds $\widetilde{\boldsymbol{x}}$ such that $\boldsymbol{A}\widetilde{\boldsymbol{x}} = \boldsymbol{b}$ and $\sum_i \boldsymbol{f}(\boldsymbol{P}\widetilde{\boldsymbol{x}}_i) \leq \sum_i \boldsymbol{f}(\boldsymbol{P}\boldsymbol{x}_i^\star) + \delta$ in at most*

$$
O\left( m^{1/3} N \mu^{-\frac{3-\nu}{2}} L^{1/2} R \log\left( \frac{\boldsymbol{f}(\boldsymbol{x}^{(0)}) - \boldsymbol{f}(\boldsymbol{x}^\star))}{\delta} \right) \right)
$$

*calls to a linear system solver.*

*Proof.* From Lemma A.1, $\boldsymbol{f}$ is $(N\mu^{-\frac{3-\nu}{2}} L^{1/2}, 2)$-g.s.c. We now use Lemma 3.3 with $M = N\mu^{-\frac{3-\nu}{2}} L^{1/2}$ followed by Theorem 4.6. $\qquad \square$

# B  Missing Proofs

## B.1  Proofs from Section 2

**Definition B.1.** *[Hessian Stability] For distance $r \in \mathbb{R}_{\geq 0}$ and function $\boldsymbol{d} : \mathbb{R}_{\geq 0} \to \mathbb{R}_{\geq 0}$ acting on $r$, a function $\boldsymbol{f}$ is $(r, \boldsymbol{d}(r))$-hessian stable w.r.t. a norm $\| \cdot \|$ if for all $\boldsymbol{x}, \boldsymbol{y}$ such that $\| \boldsymbol{x} - \boldsymbol{y} \| \leq r$,*

$$
\frac{1}{\boldsymbol{d}(r)} \nabla^2 \boldsymbol{f}(\boldsymbol{x}) \preceq \nabla^2 \boldsymbol{f}(\boldsymbol{y}) \preceq \boldsymbol{d}(r) \nabla^2 \boldsymbol{f}(\boldsymbol{x})
$$

**Lemma B.2** (Lemma 11 [CJJ$^+$20]). *If $\boldsymbol{f}$ is a univariate $M$-quasi-self-concordant (q.s.c.) function, then $\boldsymbol{f}(\boldsymbol{x}) = \sum_i \boldsymbol{f}(\boldsymbol{x}_i)$ is $(r, e^{Mr})$ hessian stable in the $\ell_\infty$-norm.*

**Lemma 2.4.** *For $\epsilon > 0$, resistances $\boldsymbol{r}$ (Definition 2.3), with corresponding weights $\boldsymbol{w}$, we have*

$$\Psi(\boldsymbol{r}) \leq (1 + \epsilon)\Phi(\boldsymbol{w}).$$

*In addition, letting $\|\boldsymbol{P}\|_{\min} = \min_{\boldsymbol{Ax} = \boldsymbol{b}} \|\boldsymbol{Px}\|_2$ and $\|\boldsymbol{A}\|$ denote the operator norm of $\boldsymbol{A}$, we have*

$$\Psi(\boldsymbol{r}) \geq \frac{\epsilon\Phi(\boldsymbol{w})}{mR^2} \frac{\|\boldsymbol{P}\|_{\min}^2 \|\boldsymbol{b}\|_2^2}{\|\boldsymbol{A}\|^2} \stackrel{\text{def}}{=} \Phi(\boldsymbol{w})L.$$

*Proof.* Let $\widetilde{\Delta}$ be the minimizer of $\Psi(\boldsymbol{r})$ and $\boldsymbol{x}^\star$ be the optimum of (1).

$$
\begin{aligned}
\Psi(\boldsymbol{r}) = \sum_i \boldsymbol{r}_i (\boldsymbol{P}\widetilde{\Delta})_i^2 &\leq \sum_i \boldsymbol{r}_i (\boldsymbol{Px}^\star)_i^2 \\
&= \sum_i \left( \boldsymbol{f}''(\boldsymbol{w}_i) + \frac{\epsilon\Phi(\boldsymbol{w})}{m} \right) \frac{(\boldsymbol{Px}^\star)_i^2}{R^2} \\
&\leq \sum_i \boldsymbol{f}''(\boldsymbol{w}_i) + \frac{\epsilon\Phi(\boldsymbol{w})}{m} \cdot m, \qquad\qquad \text{Since } \|\boldsymbol{Px}^\star\|_\infty \leq R \\
&= \Phi(\boldsymbol{w})(1 + \epsilon)
\end{aligned}
$$

We next look at a lower bound for $\Psi$. We note that, any solution to the oracle must satisfy $\boldsymbol{A}\widetilde{\Delta} = \boldsymbol{b}$. This implies, $\|\boldsymbol{A}\|\|\widetilde{\Delta}\|_2 \geq \|\boldsymbol{b}\|_2$, where $\|\cdot\|$ denotes the operator norm. Now,

$$\Psi(\boldsymbol{r}) \geq \frac{\epsilon\Phi(\boldsymbol{w})}{mR^2} \|\boldsymbol{P}\widetilde{\Delta}\|_2^2 \geq \frac{\epsilon\Phi(\boldsymbol{w})}{mR^2} \|\boldsymbol{P}\|_{\min}^2 \|\widetilde{\Delta}\|_2^2 \geq \frac{\epsilon\Phi(\boldsymbol{w})}{mR^2} \frac{\|\boldsymbol{P}\|_{\min}^2 \|\boldsymbol{b}\|_2^2}{\|\boldsymbol{A}\|^2}.$$

$\square$

**Lemma B.3.**

$$\sum_i \boldsymbol{f}''(\boldsymbol{w}_i)|\boldsymbol{P}\widetilde{\Delta}|_i \leq (1 + \epsilon)R\Phi(\boldsymbol{w})$$

*Proof.*

$$
\begin{aligned}
\sum_i \boldsymbol{f}''(\boldsymbol{w}_i)|\boldsymbol{P}\widetilde{\Delta}|_i &\leq \sqrt{\sum_i \boldsymbol{f}''(\boldsymbol{w}_i) \sum_i \boldsymbol{f}''(\boldsymbol{w}_i)|\boldsymbol{P}\widetilde{\Delta}|_i^2} \qquad \text{Cauchy Schwarz} \\
&\leq \sqrt{\Phi(\boldsymbol{w})}\sqrt{R^2\Psi(\boldsymbol{r})} \\
&\leq R\sqrt{\Phi(\boldsymbol{w})}\sqrt{(1 + \epsilon)\Phi(\boldsymbol{w})} \qquad\qquad \text{From Lemma 2.4} \\
&= R(1 + \epsilon)\Phi(\boldsymbol{w}).
\end{aligned}
$$

$\square$

## B.2 Proofs from Section 3

**Change in $\Psi$**

**Lemma 3.1.** *Let $\Psi$ be as defined in 2.3. After $t$ flow steps and $k$ width reduction steps, we have,*

$$\Psi(\boldsymbol{r}^{(t,k)}) \geq \Psi(\boldsymbol{r}^{(0,0)})\left(1 + \frac{\epsilon^2\tau^2}{(1 + \epsilon)^2 m}\right)^k \qquad \text{if } \boldsymbol{f}'' \text{ non-decreasing in } \boldsymbol{w},$$

$$\Psi(\boldsymbol{r}^{(t,k)}) \leq \Psi(\boldsymbol{r}^{(0,0)})\left(1 - \frac{\epsilon^2\tau^2}{2(1 + \epsilon)^2 m}\right)^k \qquad \text{if } \boldsymbol{f}'' \text{ non-increasing in } \boldsymbol{w}.$$

*Proof.* We show this by induction. It is clear that this holds for $t = k = 0$. We know from Lemma C.2, for $\boldsymbol{r}' \geq \boldsymbol{r}$,

$$\Psi(\boldsymbol{r}') \geq \Psi(\boldsymbol{r}) + \sum_i \left(1 - \frac{\boldsymbol{r}_i}{\boldsymbol{r}_i'}\right) \boldsymbol{r}_i (\boldsymbol{P}\widetilde{\Delta})_i^2.$$

Since the weights are only increasing, this corresponds to the case $f''$ is an increasing function. Similarly, when $f''$ is a non-increasing function, we have the following bound: for $r' \leq r$ from Lemma C.1,

$$\Psi(r') \leq \Psi(r) - \frac{1}{2} \sum_i \left(1 - \frac{r'_i}{r_i}\right) r_i (P\widetilde{\Delta})_i^2.$$

We first consider a flow step. We note that our weights $w$ are increasing, and if $f''$ is increasing then $r^{(t+1)} \geq r^{(t)}$. Similarly if $f''$ is decreasing, $r^{(t+1,k)} \leq r^{(t,k)}$. We can use the above relations to now get $\Psi(r^{(t+1,k)}) \geq \Psi(r^{(t,k)})$ for the first case and $\Psi(r^{(t+1,k)}) \leq \Psi(r^{(t,k)})$ for the second. We next consider a width reduction step. Let $i$ be one edge that has $|P\widetilde{\Delta}_i| \geq R\tau$. We have,

$$r_i^{(t,k)}(P\widetilde{\Delta})_i^2 \geq \frac{\epsilon\Phi(w^{(t,k)})}{R^2 m}|P\widetilde{\Delta}|_i^2 \geq \frac{\epsilon\Phi(w^{(t,k)})}{R^2 m}R^2\tau^2 \geq \frac{\epsilon\tau^2}{(1+\epsilon)m}\Psi(r^{(t,k)}),$$

where the last inequality follows from Lemma 2.4. Now, since we are changing our resistances by a factor of $(1 + \epsilon)$, we get the following bounds for the two cases,

$$\Psi(r^{(t,k+1)}) \geq \Psi(r^{(t,k)}) + \left(1 - \frac{r_i}{(1+\epsilon)r_i}\right)\frac{\epsilon\tau^2}{(1+\epsilon)m}\Psi(r^{(t,k)}) = \Psi(r^{(t,k)})\left(1 + \frac{\epsilon^2\tau^2}{(1+\epsilon)^2 m}\right),$$

$$\Psi(r^{(t,k+1)}) \leq \Psi(r^{(t,k)}) - \frac{1}{2}\left(1 - \frac{r_i/(1+\epsilon)}{r_i}\right)\frac{\epsilon\tau^2}{(1+\epsilon)m}\Psi(r^{(t,k)}) = \Psi(r^{(t,k)})\left(1 - \frac{\epsilon^2\tau^2}{2(1+\epsilon)^2 m}\right).$$

With these two relations we conclude our proof. $\qquad\square$

**Change in $\Phi$**

**Lemma 3.2.** *Suppose $f$ is $M$-q.s.c. Let $\alpha$ and $\tau$ be such that $\alpha\tau \leq M^{-1}$. After $t$ flow steps and $k$ width reduction steps, our potential $\Phi$ satisfies*

$$\Phi(w^{(t,k)}) \leq \left(1 + \epsilon(1+\epsilon)^2\alpha M\right)^t \left(1 + \epsilon(1+\epsilon)\tau^{-1}\right)^k \Phi(w_0) \quad \text{if } f'' \text{ non-decreasing in } w,$$

$$\Phi(w^{(t,k)}) \geq \left(1 - \epsilon(1+\epsilon)^2\alpha M\right)^t \left(1 - \epsilon(1+\epsilon)\tau^{-1}\right)^k \Phi(w_0) \quad \text{if } f'' \text{ non-increasing in } w.$$

*Proof.* We first show the case when $f''$ is increasing. The same calculation will work for the other case too by just considering the sign of $\Phi'$.

We will use induction. It is easy to see the claim holds for the initial iteration, $t = k = 0$. We next assume that it holds for some $w^{(t,k)}$. If the next step is a flow step, we update to $w^{(t+1,k)} \leq w^{(t,k)} + \epsilon\alpha\tau$. Since $\alpha\tau \leq M^{-1}$, we have that $\Phi$ is $(M^{-1}, e^\epsilon)$ hessian stable around this update. We will use $w$ to denote $w^{(t,k)}$ for simplicity. We thus have,

$$
\begin{aligned}
\Phi(w^{(t+1)}) =& \Phi\left(w + \frac{\epsilon\alpha}{R}|P\widetilde{\Delta}|\right) \\
=& \Phi(w) + \frac{\epsilon\alpha}{R}\nabla\Phi(y)^\top|P\widetilde{\Delta}| \\
& \text{(For some } y \text{ between } w \text{ and } w + \alpha|P\Delta|) \\
=& \Phi(w) + \frac{\epsilon\alpha}{R}\sum_i f'''(y_i)|P\widetilde{\Delta}|_i \\
\leq& \Phi(w) + \frac{\epsilon\alpha}{R}M\sum_i f''(y_i)|P\widetilde{\Delta}|_i \\
& \text{(Since } f \text{ is } M\text{-q.s.c.)} \\
\leq& \Phi(w) + \frac{\epsilon\alpha}{R}Me^\epsilon\sum_i f''(w_i)|P\widetilde{\Delta}|_i \\
& \text{(Since } f \text{ is hessian stable in this range)} \\
\leq& \Phi(w) + \epsilon(1+\epsilon)^2\alpha M\Phi(w) \\
& \text{(From Lemma B.3)}
\end{aligned}
$$

We thus get the following bound,

$$\Phi(\boldsymbol{w}^{(t+1,k)}) \le \Phi(\boldsymbol{w}^{(t,k)})\Big(1 + \epsilon(1+\epsilon)^2 \alpha M\Big).$$

Now, suppose the next step is a width reduction step.

$$
\begin{aligned}
\Phi(\boldsymbol{w}^{(t,k+1)}) &= \sum_{i \notin \mathcal{I}} \Phi(\boldsymbol{w}_i) + \sum_{i \in \mathcal{I}} \Phi\Big(\boldsymbol{w}_i^{(t+1)}\Big) \\
&= \sum_{i \notin \mathcal{I}} \Phi(\boldsymbol{w}_i) + \sum_{i \in \mathcal{I}} \boldsymbol{f}''\Big(\boldsymbol{w}_i^{(t+1)}\Big) \\
&\le \sum_{i \notin \mathcal{I}} \Phi(\boldsymbol{w}_i) + (1+\epsilon) \sum_{i \in \mathcal{I}} \boldsymbol{f}''(\boldsymbol{w}_i) \\
&\le \Phi(\boldsymbol{w}) + \frac{\epsilon}{R\tau} \sum_{i \in \mathcal{I}} \boldsymbol{f}''(\boldsymbol{w}_i)|\boldsymbol{P}\widetilde{\Delta}|_i \\
&\le \Phi(\boldsymbol{w}) + \frac{\epsilon}{R\tau} \sum_{i} \boldsymbol{f}''(\boldsymbol{w}_i)|\boldsymbol{P}\widetilde{\Delta}|_i \\
&\le \Phi(\boldsymbol{w}) + \frac{\epsilon(1+\epsilon)}{\tau}\Phi(\boldsymbol{w}) \\
&\qquad\qquad \text{From Lemma B.3}
\end{aligned}
$$

We thus get the following bound,

$$\Phi(\boldsymbol{w}^{(t,k+1)}) \le \Phi(\boldsymbol{w}^{(t,k)})\Big(1 + \epsilon(1+\epsilon)\tau^{-1}\Big).$$

$\square$

## B.3    Proofs from Section 4

### Iterative Refinement

**Lemma B.4.** *Let $\boldsymbol{f}$ be a $(r, d(r))$-hessian stable function in $\ell_\infty$-norm, and $\widetilde{\boldsymbol{x}} = \boldsymbol{x} + \Delta$ such that $\|\Delta\|_\infty \le r$. We then have,*

$$\frac{1}{d(r)}\Delta^\top \nabla^2 \boldsymbol{f}(\boldsymbol{x})\Delta \le \boldsymbol{f}(\widetilde{\boldsymbol{x}}) - \boldsymbol{f}(\boldsymbol{x}) - \nabla \boldsymbol{f}(\boldsymbol{x})^\top \Delta \le d(r)\Delta^\top \nabla^2 \boldsymbol{f}(\boldsymbol{x})\Delta,$$

*Proof.* We have for some $\boldsymbol{z}$ along the line joining $\boldsymbol{x}$ and $\widetilde{\boldsymbol{x}}$,

$$\boldsymbol{f}(\widetilde{\boldsymbol{x}}) = \boldsymbol{f}(\boldsymbol{x}) + \nabla \boldsymbol{f}(\boldsymbol{x})^\top \Delta + \Delta^\top \nabla^2 \boldsymbol{f}(\boldsymbol{z})\Delta.$$

Since $\|\boldsymbol{z} - \boldsymbol{x}\|_\infty \le \|\widetilde{\boldsymbol{x}} - \boldsymbol{x}\|_\infty \le r$, from hessian stability, we have,

$$\frac{1}{d(r)}\nabla^2 \boldsymbol{f}(\boldsymbol{x}) \preceq \nabla^2 \boldsymbol{f}(\boldsymbol{z}) \preceq d(r)\nabla^2 \boldsymbol{f}(\boldsymbol{x}).$$

Using this relation in the above, we get our lemma. $\square$

**Lemma B.5.** *Let $\Delta$ be any feasible solution to the residual problem at $\boldsymbol{x}$. We then have,*

$$\boldsymbol{f}(\boldsymbol{x}) - \boldsymbol{f}(\boldsymbol{x} - \Delta) \le res(\Delta), \quad \boldsymbol{f}(\boldsymbol{x}) - \boldsymbol{f}(\boldsymbol{x} - e^{-2}\Delta) \ge e^{-2} \cdot res(\Delta),$$

*Proof.* Since our function is $M$-q.s.c., from Lemmas B.4 and B.2, for all $\Delta$ such that $\|\boldsymbol{P}\Delta\|_\infty \le M^{-1}$,

$$e^{-1}(\boldsymbol{P}\Delta)^\top \nabla^2 \boldsymbol{f}(\boldsymbol{x})\boldsymbol{P}\Delta \le \boldsymbol{f}(\boldsymbol{x} - \Delta) - \boldsymbol{f}(\boldsymbol{x}) + \nabla \boldsymbol{f}(\boldsymbol{x})^\top \boldsymbol{P}\Delta \le e(\boldsymbol{P}\Delta)^\top \nabla^2 \boldsymbol{f}(\boldsymbol{x})\boldsymbol{P}\Delta.$$

The first bound directly follows from the left inequality. For the second bound, we first note that $e^{-2}\|\boldsymbol{P}\Delta\| \le M^{-1}$. We can now use the right inequality.

$$\boldsymbol{f}(\boldsymbol{x}) - \boldsymbol{f}(\boldsymbol{x} - e^{-2}\Delta) \ge e^{-2}\nabla\boldsymbol{f}(\boldsymbol{x})^\top\boldsymbol{P}\Delta - e^{-3}(\boldsymbol{P}\Delta)^\top\nabla^2\boldsymbol{f}(\boldsymbol{x})\boldsymbol{P}\Delta$$
$$= e^{-2}\Big(\nabla\boldsymbol{f}(\boldsymbol{x})^\top\boldsymbol{P}\Delta - e^{-1}(\boldsymbol{P}\Delta)^\top\nabla^2\boldsymbol{f}(\boldsymbol{x})\boldsymbol{P}\Delta\Big)$$
$$= e^{-2}res(\Delta).$$

$\square$

**Lemma B.6.** *Assume $\boldsymbol{f}$ is $M$-q.s.c. Let $\boldsymbol{x}^\star$ denote the minimizer of Problem (1) and $\Delta^\star$ the optimizer of Problem (3) at $\boldsymbol{x}^{(t)}$. We then have,*

$$res(\Delta^\star) \ge \frac{1}{4MR}\Big(\boldsymbol{f}(\boldsymbol{x}^{(t)}) - \boldsymbol{f}(\boldsymbol{x}^\star)\Big).$$

*Proof.* Let $\boldsymbol{x}^{(t)}$ be such that $\boldsymbol{A}\boldsymbol{x}^{(t)} = \boldsymbol{b}$ and $\boldsymbol{x}^\star$ is the optimum of (1). Note that we have $\|\boldsymbol{P}\boldsymbol{x}^{(t)}\|_\infty \le R$ and therefore, $\left\|\boldsymbol{P}\boldsymbol{x}^{(t)} - \boldsymbol{P}\boldsymbol{x}^\star\right\|_\infty \le 2R$. Let $r = \frac{1}{2M}$ and $\boldsymbol{x} = \left(1 - \frac{r}{2R}\right)\boldsymbol{x}^{(t)} + \frac{r}{2R}\boldsymbol{x}^\star$. Let $\widetilde{\Delta} = \boldsymbol{x}^{(t)} - \boldsymbol{x} = \frac{r}{2R}(\boldsymbol{x}^{(t)} - \boldsymbol{x}^\star)$. We have,

$$\left\|\boldsymbol{P}\widetilde{\Delta}\right\|_\infty = \left\|\boldsymbol{P}\boldsymbol{x}^{(t)} - \boldsymbol{P}\boldsymbol{x}\right\|_\infty = \frac{r}{2R}\left\|\boldsymbol{P}\boldsymbol{x}^{(t)} - \boldsymbol{P}\boldsymbol{x}^\star\right\|_\infty \le r,$$

and

$$\boldsymbol{A}\widetilde{\Delta} = \boldsymbol{A}(\boldsymbol{x}^{(t)} - \boldsymbol{x}) = \frac{r}{2R}(-\boldsymbol{A}\boldsymbol{x}^\star + \boldsymbol{A}\boldsymbol{x}^{(t)}) = 0.$$

We next show that $\left\|\boldsymbol{P}\widetilde{\Delta} - \boldsymbol{z}\right\|_\infty \le \frac{1}{2M}$.

$$\left\|\boldsymbol{P}\widetilde{\Delta} - \boldsymbol{z}\right\|_\infty = \left\|\frac{r}{2R}\boldsymbol{P}\boldsymbol{x}^{(t)} - \frac{r}{2R}\boldsymbol{P}\boldsymbol{x}^\star - \boldsymbol{z}\right\|_\infty$$

We will do a case by case analysis. Consider some coordinate $i$.

1. $\boldsymbol{P}\boldsymbol{x}_i^{(t)} - \frac{1}{2M} < -R$: From the definition of $\boldsymbol{z}_i$, we note that $\boldsymbol{z}_i = R - \frac{1}{2M} + \boldsymbol{P}\boldsymbol{x}_i^{(t)}$ and $-R < \boldsymbol{P}\boldsymbol{x}_i^{(t)} \le -R + \frac{1}{2M}$. Suppose $\boldsymbol{P}\boldsymbol{x}_i^{(t)} = -R + a$ for some $0 \le a < \frac{1}{2M}$. We have,

$$\left|\boldsymbol{P}\widetilde{\Delta} - \boldsymbol{z}\right|_i = \left|\frac{r}{2R}(\boldsymbol{P}\boldsymbol{x}_i^{(t)} - \boldsymbol{P}\boldsymbol{x}_i^\star) - \boldsymbol{z}_i\right|$$
$$= \left|\frac{r}{2R}(-R + a - \boldsymbol{P}\boldsymbol{x}_i^\star) - a + \frac{1}{2M}\right|$$
$$= \left|\frac{r}{2R}(-R - \boldsymbol{P}\boldsymbol{x}_i^\star) - a\left(1 - \frac{r}{2R}\right) + \frac{1}{2M}\right|$$
$$\le \frac{1}{2M}.$$

The last inequality follows since $-2R \le -R - \boldsymbol{P}\boldsymbol{x}_i^\star \le 0$.

2. $\boldsymbol{P}\boldsymbol{x}_i^{(t)} + \frac{1}{2M} > R$: From the definition of $\boldsymbol{z}_i$, we note that $\boldsymbol{z}_i = -R + \frac{1}{2M} + \boldsymbol{P}\boldsymbol{x}_i^{(t)}$ and $R - \frac{1}{2M} < \boldsymbol{P}\boldsymbol{x}_i^{(t)} \le R$. Suppose $\boldsymbol{P}\boldsymbol{x}_i^{(t)} = R - a$ for some $0 \le a < \frac{1}{2M}$. We have,

$$\left|\boldsymbol{P}\widetilde{\Delta} - \boldsymbol{z}\right|_i = \left|\frac{r}{2R}(\boldsymbol{P}\boldsymbol{x}_i^{(t)} - \boldsymbol{P}\boldsymbol{x}_i^\star) - \boldsymbol{z}_i\right|$$
$$= \left|\frac{r}{2R}(R - a - \boldsymbol{P}\boldsymbol{x}_i^\star) + a - \frac{1}{2M}\right|$$
$$= \left|\frac{r}{2R}(R - \boldsymbol{P}\boldsymbol{x}_i^\star) + a\left(1 - \frac{r}{2R}\right) - \frac{1}{2M}\right|$$
$$\le \frac{1}{2M}.$$

The last inequality follows since $0 \le R - \boldsymbol{P}\boldsymbol{x}_i^\star \le 2R$.

3. $-R + \frac{1}{2M} \le \boldsymbol{P}\boldsymbol{x}_i^{(t)} \le -\frac{1}{2M}R$: In this case $\boldsymbol{z}_i = 0$.

$$\left|\boldsymbol{P}\widetilde{\Delta} - \boldsymbol{z}\right|_i = \left|\frac{r}{2R}(\boldsymbol{P}\boldsymbol{x}_i^{(t)} - \boldsymbol{P}\boldsymbol{x}_i^\star)\right| \le r = \frac{1}{2M}.$$

We thus conclude, that $\boldsymbol{x} - \boldsymbol{x}^{(t)}$ is a feasible solution for the residual problem and from convexity,

$$\frac{r}{2R}\Big(\boldsymbol{f}(\boldsymbol{x}^{(t)}) - \boldsymbol{f}(\boldsymbol{x}^\star)\Big) \le \boldsymbol{f}(\boldsymbol{x}^{(t)}) - \boldsymbol{f}(\boldsymbol{x}).$$

Let $\Delta^\star$ denote the optimum of the residual problem at $\boldsymbol{x}^{(t)}$ (3). From Lemma B.5,

$$\frac{r}{2R}\Big(\boldsymbol{f}(\boldsymbol{x}^{(t)}) - \boldsymbol{f}(\boldsymbol{x}^\star)\Big) \le \boldsymbol{f}(\boldsymbol{x}^{(t)}) - \boldsymbol{f}(\boldsymbol{x}) \le res\Big(\boldsymbol{x}^{(t)} - \boldsymbol{x}\Big) \le res(\Delta^\star).$$

$\square$

**Lemma 4.2.** *[Iterative Refinement] Let $\boldsymbol{f}$ be $M$-q.s.c. and $\widetilde{\Delta}^{(t)}$ a $\kappa$-approximate solution to the residual problem at $\boldsymbol{x}^{(t)}$ (Problem (3)). Starting from $\boldsymbol{x}^{(0)}$ such that $\boldsymbol{A}\boldsymbol{x}^{(0)} = \boldsymbol{b}$, $\|\boldsymbol{x}^{(0)}\|_\infty \le R$, and iterating as $\boldsymbol{x}^{(t+1)} = \boldsymbol{x}^{(t)} - e^{-2}\widetilde{\Delta}^{(t)}$, after at most $O\left(\kappa M R \log\left(\frac{\boldsymbol{f}(\boldsymbol{x}^{(0)}) - \boldsymbol{f}(\boldsymbol{x}^\star)}{\epsilon}\right)\right)$ iterations we get $\boldsymbol{x}$ such that $\boldsymbol{A}\boldsymbol{x} = \boldsymbol{b}$ and $\boldsymbol{f}(\boldsymbol{x}) \le \boldsymbol{f}(\boldsymbol{x}^\star) + \epsilon$.*

*Proof.* From Lemma B.6,

$$res(\widetilde{\Delta}^{(t)}) \ge \frac{1}{\kappa}res(\Delta^\star) \ge \frac{1}{4\kappa M R}\Big(\boldsymbol{f}(\boldsymbol{x}^{(t)}) - \boldsymbol{f}(\boldsymbol{x}^\star)\Big).$$

Now, from Lemma B.5,

$$\boldsymbol{f}(\boldsymbol{x}^{(t+1)}) - \boldsymbol{f}(\boldsymbol{x}^\star) \le \boldsymbol{f}(\boldsymbol{x}^{(t)}) - \boldsymbol{f}(\boldsymbol{x}^\star) - e^{-2}res(\widetilde{\Delta}^{(t)}) \le \left(1 - \frac{e^{-2}}{4\kappa M R}\right)\Big(\boldsymbol{f}(\boldsymbol{x}^{(t)}) - \boldsymbol{f}(\boldsymbol{x}^\star)\Big).$$

Inductively applying the above equation,

$$\boldsymbol{f}(\boldsymbol{x}^{(T)}) - \boldsymbol{f}(\boldsymbol{x}^\star) \le \left(1 - \frac{e^{-2}}{4\kappa M R}\right)^T \Big(\boldsymbol{f}(\boldsymbol{x}^{(0)}) - \boldsymbol{f}(\boldsymbol{x}^\star)\Big).$$

$\square$

### Binary Search

**Lemma 4.3.** *Let $\nu$ be such that $\boldsymbol{f}(\boldsymbol{x}^{(t)}) - \boldsymbol{f}(\boldsymbol{x}^\star) \in (\nu/2, \nu]$ and $\Delta^\star$ denote the optimum of the residual problem at $\boldsymbol{x}^{(t)}$. Then, $res(\Delta^\star) \in \left(\frac{\nu}{8MR}, e^2\nu\right]$.*

*Proof.* The lower bound follows form B.6. For the upper bound, from B.5,

$$\nu \ge \boldsymbol{f}(\boldsymbol{x}^{(t)}) - \boldsymbol{f}(\boldsymbol{x}^\star) \ge \boldsymbol{f}(\boldsymbol{x}^{(t)}) - \boldsymbol{f}(\boldsymbol{x} - e^{-2}\Delta^\star) \ge e^{-2}res(\Delta^\star).$$

$\square$

**Lemma 4.4.** *Let $\zeta$ be such that $res(\Delta^\star) \in (\zeta/2, \zeta]$ and $\Delta^\star$ the optimum of the residual problem. Then, $(\boldsymbol{P}\Delta^\star)^\top \nabla^2 \boldsymbol{f}(\boldsymbol{x})\boldsymbol{P}\Delta^\star \le e \cdot \zeta$.*

*Proof.* Consider scaling $\Delta^\star$ by $O(1) > \lambda > 0$. We must have,

$$\left[\frac{d}{d\lambda}res(\lambda\Delta^\star)\right]_{\lambda=1} = 0.$$

This implies,

$$\nabla \boldsymbol{f}(\boldsymbol{x})^\top \boldsymbol{P}\Delta^\star - 2e^{-1}(\boldsymbol{P}\Delta^\star)^\top \nabla^2 \boldsymbol{f}(\boldsymbol{x})\boldsymbol{P}\Delta^\star = 0,$$

or

$$e^{-1}(\boldsymbol{P}\Delta^\star)^\top \nabla^2 \boldsymbol{f}(\boldsymbol{x})\boldsymbol{P}\Delta^\star = \nabla \boldsymbol{f}(\boldsymbol{x})^\top \boldsymbol{P}\Delta^\star - e^{-1}(\boldsymbol{P}\Delta^\star)^\top \nabla^2 \boldsymbol{f}(\boldsymbol{x})\boldsymbol{P}\Delta^\star = res(\Delta^\star) \le \zeta.$$

$\square$

**Width Reduction**

**Lemma 4.5.** *Let $\zeta$ be such that $res(\Delta^\star) \in (\zeta/2, \zeta]$. Algorithm 3 returns $\boldsymbol{y}$ such that $\boldsymbol{Ay} = 0$, $\|\boldsymbol{Py} - \boldsymbol{z}\|_\infty \leq \frac{1}{2M}$ and $res(\boldsymbol{y}) \geq \frac{1}{400} res(\Delta^\star)$ in $O(m^{1/3})$ calls to a linear system solver.*

*Proof.* This algorithm is basically an implementation of the width-reduced MWU algorithm from [CKM+11]. We will give a proof for completeness. For the purpose of this proof, we denote,

$$\Psi(\boldsymbol{r}) = \min_{\boldsymbol{A\Delta}=0, \nabla \boldsymbol{f}(\boldsymbol{x})^\top \boldsymbol{P\Delta} = \zeta/2} \sum_j \left( \boldsymbol{f}''(\boldsymbol{x}_j)(\boldsymbol{P\Delta})_j^2 + \sum_j 4M^2 \left( \boldsymbol{w}_j + \frac{\|\boldsymbol{w}\|_1}{m} \right) \right) (\boldsymbol{P\Delta} - \boldsymbol{z})_j^2,$$

$$\Phi(\boldsymbol{w}) = \|\boldsymbol{w}\|_1.$$

Let $\widetilde{\Delta}$ be the solution returned by $\Psi$. We first note that, for $\Delta^\star$ the optimum of the residual problem,

$$\Psi(\boldsymbol{r}) \leq \sum_j \left( \boldsymbol{f}''(\boldsymbol{x}_j)(\boldsymbol{P\Delta}^\star)_j^2 + \sum_j 4M^2 \left( \boldsymbol{w}_j + \frac{\|\boldsymbol{w}\|_1}{m} \right) \right) (\boldsymbol{P\Delta}^\star - \boldsymbol{z})_j^2$$

$$\leq e \cdot \zeta + \sum_j 4M^2 \left( \boldsymbol{w}_j + \frac{\|\boldsymbol{w}\|_1}{m} \right) (\boldsymbol{P\Delta}^\star - \boldsymbol{z})_j^2, \text{ From Lemma 4.4}$$

$$\leq e \cdot \zeta + \|\boldsymbol{w}\|_1 + \Phi(\boldsymbol{w}), \text{ Since } \|\boldsymbol{P\Delta}^\star - \boldsymbol{z}\|_\infty \leq \frac{1}{2M}$$

$$\leq (e + 2)\Phi(\boldsymbol{w}).$$

We note that,

$$\sum_j \boldsymbol{w}_j (4M)(\boldsymbol{P\widetilde{\Delta}} - \boldsymbol{z})_j \leq \sqrt{\sum_j \boldsymbol{w}_j \sum_j \boldsymbol{w}_j (4M)^2 (\boldsymbol{P\widetilde{\Delta}} - \boldsymbol{z})_j^2} \leq \sqrt{\Phi(\boldsymbol{w})\Psi(\boldsymbol{r})} \leq \sqrt{e + 2}\Phi(\boldsymbol{w}).$$

(4)

For a flow step, from the above calculation, note that,

$$\Phi(\boldsymbol{w}^{(t+1)}) = \sum_j \boldsymbol{w}_j + \frac{\alpha}{2} \sum_j \boldsymbol{w}_j M(\boldsymbol{P\widetilde{\Delta}} - \boldsymbol{z})_j \leq \Phi(\boldsymbol{w}^{(t)}) + \frac{\sqrt{e+2}}{8} \alpha \Phi(\boldsymbol{w}^{(t)}) = \Phi(\boldsymbol{w}^{(t)})(1 + \alpha).$$

For a width reduction step let $\mathcal{I}$ denote the indices which have the weights doubled,

$$\Phi(\boldsymbol{w}^{(t+1)}) = \sum_{j \notin \mathcal{I}} \boldsymbol{w}_j^{(t)} + 2 \sum_{j \in \mathcal{I}} \boldsymbol{w}_j^{(t)} \leq \Phi(\boldsymbol{w}^{(t)}) + \frac{2}{\tau} \sum_{j \in \mathcal{I}} \boldsymbol{w}_j^{(t)} (2M)|\boldsymbol{P\widetilde{\Delta}} - \boldsymbol{z}|_j$$

$$\leq \Phi(\boldsymbol{w}^{(t)}) + \frac{\sqrt{e+2}}{\tau} \Phi(\boldsymbol{w}) \leq \Phi(\boldsymbol{w}^{(t)}\left(1 + 3\tau^{-1}\right).$$

We can bound the number of width reduction steps by $O(m/\tau^2)$ similar to Lemma 3.1. We now show that our final solution has $\|\frac{1}{T}\boldsymbol{Py} - \boldsymbol{z}\|_\infty \leq \frac{1}{2M}$. After $T$ iterations, let $j$ denote the index with max value in vector $\boldsymbol{w}$. For $\alpha\tau \leq 1$, $\left(1 + \frac{\alpha}{2}M|\boldsymbol{P\widetilde{\Delta}} - \boldsymbol{z}|_j\right) \geq \exp\left(\frac{3}{4}\alpha M|\boldsymbol{P\widetilde{\Delta}} - \boldsymbol{z}|_j\right)$.

$$10\zeta \geq \Phi(\boldsymbol{w}^T) \geq \boldsymbol{w}_j^{(T)} \geq \frac{\zeta}{m} \Pi_{t=1}^T \left(1 + \frac{\alpha}{2}M|\boldsymbol{P\widetilde{\Delta}}^{(t)} - \boldsymbol{z}|_j\right)$$

$$\geq \frac{\zeta}{m} \exp\left(\frac{3}{8}\alpha(2M) \sum_t |\boldsymbol{P\widetilde{\Delta}}^{(t)} - \boldsymbol{z}|_j\right) = \frac{\zeta}{m} \exp\left(\frac{3}{8}\alpha(2M)(\boldsymbol{Py} - T\boldsymbol{z})_j\right).$$

We thus have for all coordinates $j$ and $T \geq \alpha^{-1} O(\log m)$,

$$\frac{|\boldsymbol{Py} - T\boldsymbol{z}|_j}{T} \leq \frac{O(M^{-1}\log m)}{\alpha T} \leq \frac{1}{2M}.$$

It remains to show that $\boldsymbol{y}/(100T)$ has the required value for the residual. First note that,

$$\nabla \boldsymbol{f}(\boldsymbol{x})^\top \frac{\boldsymbol{y}}{100T} = \frac{1}{100T} \sum_t \nabla \boldsymbol{f}(\boldsymbol{x})^\top \boldsymbol{P\widetilde{\Delta}}^{(t)} = \frac{\zeta}{2 \cdot 100}.$$

We next look at the quadratic term.

$$\frac{1}{(100)^2 T^2}\sum_j \boldsymbol{f}''(\boldsymbol{x}_j)\boldsymbol{y}_j^2 = \frac{1}{T^2(100)^2}\sum_j \boldsymbol{f}''(\boldsymbol{x}_j)\left(\sum_t |\boldsymbol{P}\widetilde{\Delta}^{(t)}|_j\right)^2$$

$$\leq \frac{1}{T^2(100)^2}\sum_j T\sum_t \boldsymbol{f}''(\boldsymbol{x}_j)|\boldsymbol{P}\widetilde{\Delta}^{(t)}|_j^2 = \frac{1}{T(100)^2}\sum_t \Psi(\boldsymbol{r}^{(t)})$$

$$\leq \frac{1}{T(100)^2}T(e+2)\Phi(\boldsymbol{w}^{(T)}) \leq \frac{10(e+2)}{(100)^2}\zeta.$$

Choose $c$ such that we have,

$$e^{-1}\frac{1}{(100)^2}\sum_j \boldsymbol{f}''(\boldsymbol{x}_j)\boldsymbol{y}_j^2 \leq \frac{\zeta}{4\cdot 100}.$$

We thus have,

$$res\left(\frac{\boldsymbol{y}}{100T}\right) = \nabla\boldsymbol{f}(\boldsymbol{x})^\top \frac{\boldsymbol{y}}{100T} - e^{-1}\frac{1}{(100)^2 T^2}\sum_j \boldsymbol{f}''(\boldsymbol{x}_j)\boldsymbol{y}_j^2 \geq \frac{\zeta}{4\cdot 100} \geq \frac{1}{400}res(\Delta^\star).$$

$\square$

## B.4 Proofs from Section 5

**Sum of exponential, soft-max and $\ell_\infty$ regression**

**Theorem 5.2.** *Let $\boldsymbol{x}^\star$ denote the optimum of the $\ell_\infty$-regression problem, $\min_{\boldsymbol{A}\boldsymbol{x}=\boldsymbol{b}}\|\boldsymbol{P}\boldsymbol{x}\|_\infty$. Algorithm 1 when applied to the function $\boldsymbol{f}(\boldsymbol{P}\boldsymbol{x}) = \sum_i \left(e^{\frac{(\boldsymbol{P}\boldsymbol{x})_i}{\nu}} + e^{\frac{-(\boldsymbol{P}\boldsymbol{x})_i}{\nu}}\right)$ for $\nu = \Omega\left(\frac{\epsilon}{\log m}\right)$, returns $\widetilde{\boldsymbol{x}}$ such that $\boldsymbol{A}\widetilde{\boldsymbol{x}} = \boldsymbol{b}$ and*

$$\|\boldsymbol{P}\widetilde{\boldsymbol{x}}\|_\infty \leq (1+\epsilon)\|\boldsymbol{P}\boldsymbol{x}^\star\|_\infty,$$

*in at most $\widetilde{O}(m^{1/3}\epsilon^{-5/3})$ calls to a linear system solve.*

*Proof.* Let $\boldsymbol{Q} = \begin{bmatrix} P \\ -P \end{bmatrix}$. We note that $\boldsymbol{f}(\boldsymbol{x}) = \sum_i e^{\frac{(\boldsymbol{Q}\boldsymbol{x})_i}{\nu}}$. Let $\overline{\boldsymbol{x}}$ denote the optimum of $\boldsymbol{f}$, which is also the optimum of $smax_\nu(\boldsymbol{Q}\boldsymbol{x})$. We have the following relation,

$$\forall \boldsymbol{x}, \|\boldsymbol{P}\boldsymbol{x}\|_\infty \leq smax_\nu(\boldsymbol{Q}\boldsymbol{x}) \leq \|\boldsymbol{P}\boldsymbol{x}\|_\infty + \nu\log m.$$

Let $R = \|\boldsymbol{P}\boldsymbol{x}^\star\|_\infty$ (we can find this up to $\epsilon$ error using binary search), then the above relation implies $smax_\nu(\boldsymbol{Q}\overline{\boldsymbol{x}}) \leq R(1+\epsilon)$. From Theorem 5.1,

$$\|\boldsymbol{P}\widetilde{\boldsymbol{x}}\|_\infty \leq smax_\nu(\boldsymbol{Q}\widetilde{\boldsymbol{x}}) \leq R(1+\epsilon) = \|\boldsymbol{P}\boldsymbol{x}^\star\|_\infty(1+\epsilon). \qquad \square$$

**Theorem 5.3.** *For $\delta > 0$, let $\overline{\boldsymbol{x}}$ be the solution returned by Algorithm 1 (with $\epsilon = 1$) applied to $\boldsymbol{f}(\boldsymbol{P}\boldsymbol{x}) = \sum_i e^{\frac{(\boldsymbol{P}\boldsymbol{x})_i}{\nu}}$. Now, Algorithm 2 with starting solution $\boldsymbol{x}^{(0)} = \overline{\boldsymbol{x}}$, applied to $\boldsymbol{f}$ finds $\widetilde{\boldsymbol{x}}$ such that $\boldsymbol{A}\widetilde{\boldsymbol{x}} = \boldsymbol{b}$ and $\sum_i e^{\frac{(\boldsymbol{P}\widetilde{\boldsymbol{x}})_i}{\nu}} \leq (1+\delta)\sum_i e^{\frac{(\boldsymbol{P}\boldsymbol{x}^\star)_i}{\nu}}$ in at most $O\left(m^{1/3}R^2\nu^{-2}\log\left(\frac{m}{\delta}\right)\right)$ calls to a linear system solver.*

*Proof.* From Lemma 3.3, Algorithm 1 returns $\overline{\boldsymbol{x}}$ in $O(m^{1/3})$ iterations such that $\boldsymbol{A}\overline{\boldsymbol{x}} = \boldsymbol{b}$ and $\|\boldsymbol{P}\overline{\boldsymbol{x}}\|_\infty \leq MR\|\boldsymbol{w}^{(T,K)}\|_\infty$. Since $\frac{1}{\nu^2}\sum_i e^{\frac{w_i^{(T,K)}}{\nu}} = \Phi(\boldsymbol{w}^{(T,K)}) \leq \Phi(\boldsymbol{w}_0)e^5$, we have $\|\boldsymbol{w}^{(T,K)}\|_\infty \leq 5\nu$. This gives, $\|\boldsymbol{P}\overline{\boldsymbol{x}}\|_\infty \leq 5R$. We next bound the function value.

$$\boldsymbol{f}(\boldsymbol{P}\overline{\boldsymbol{x}}) = \sum_i e^{\frac{\boldsymbol{P}\boldsymbol{x}_i}{\nu}} \leq \sum_i e^{\frac{w_i^{(T,K)}MR}{\nu}}.$$

If $MR \leq 1$, then $\boldsymbol{f}(\boldsymbol{P}\overline{\boldsymbol{x}}) \leq \nu^2 \Phi(\boldsymbol{w}^{(T,K)}) \leq m$. Otherwise,

$$\boldsymbol{f}(\boldsymbol{P}\overline{\boldsymbol{x}}) \leq \sum_i \left(e^{\frac{w_i^{(T,K)}}{\nu}}\right)^{MR} \leq \left(\sum_i e^{\frac{w_i^{(T,K)}}{\nu}}\right)^{MR} \leq (\nu^2\Phi(\boldsymbol{w}^{(T,K)}))^{MR} \leq O(m^{MR}).$$

Now, we use Algorithm 2. Using the above calculated bounds in Theorem 4.6 we get our result. $\square$

**$\ell_p$-Regression**

**Theorem 5.4.** *For $\delta > 0$ and $p \geq 3$, let $\overline{x}$ be the solution returned by Algorithm 1 (with $\epsilon = 1$) applied to $f(Px) = \|Px\|_p^p + \mu\|Px\|_2^2$. Now, Algorithm 2 with starting solution $x^{(0)} = \overline{x}$, applied to $f$ finds $\widetilde{x}$ such that $A\widetilde{x} = b$ and $f(P\widetilde{x}) \leq f(Px^\star) + \delta$ in at most $O\left( p^2 \mu^{-1/(p-2)} m^{1/3} R \log\left( \frac{pmR}{\mu\delta} \right) \right)$ calls to a linear system solver.*

*Proof.* From Lemma 3.3, we get $\overline{x}$ such that $\|\overline{x}\|_\infty \leq RM\|w^{(T,K)}\|_\infty$. We now want to bound $f(\overline{x})$.

$$f(\overline{x}) = (RM)^p \|w^{(T,K)}\|_p^p + \mu(RM)^2 \|w^{(T,K)}\|_2^2.$$

We next note that for $w^{(T,K)} \geq w_0 = 1$,

$$\Phi(w^{(T,K)}) = p(p-1)\|w^{(T,K)}\|_{p-2}^{p-2} + 2\mu \leq \Phi(w_0)e^{O(1)}.$$

This implies that $w^{(T,K)} \leq O(1)w_0$ and $\|w^{(T,K)}\|_\infty \leq O(1)$. Therefore,

$$f(\overline{x}) \leq \big((O(1)RM\big)^p m.$$

Now, using this bound on $f(\overline{x})$ and $\overline{x}$ as a starting solution for Algorithm 2, we get our result by applying Theorem 4.6. $\qquad\square$

### B.4.1 Logistic Regression

**Theorem 5.5.** *For $\delta > 0$, let $\overline{x}$ be the solution returned by Algorithm 1 (with $\epsilon = 1$) applied to $f(Px) = \sum_i \log(1 + e^{(Px)_i})$. Now, Algorithm 2 with starting solution $x^{(0)} = \overline{x}$, applied to $f$ finds $\widetilde{x}$ such that $A\widetilde{x} = b$ and $\sum_i \log(1 + e^{(P\widetilde{x})_i}) \leq \sum_i \log(1 + e^{(Px^\star)_i}) + \delta$ in at most $O\left( m^{1/3} R \log\left( \frac{mR}{\delta} \right) \right)$ calls to a linear system solver.*

*Proof.* From Lemma 3.3, we get $\overline{x}$ such that $\|\overline{x}\|_\infty \leq RM\|w^{(T,K)}\|_\infty$. We now want to bound $f(\overline{x})$.

$$f(\overline{x}) = \sum_i \log(1 + e^{RM w_i^{(T,K)}}) \leq 2RM \sum_i w_i^{(T,K)}.$$

We next note that for $w^{(T,K)} \geq w_0$,

$$\Phi(w^{(T,K)}) = \sum_i \frac{e^{w_i^{(T,K)}}}{(1 + e^{w_i^{(T,K)}})^2} \geq \Phi(w_0)e^{-O(1)}.$$

This implies that $w^{(T,K)} \leq O(1)w_0$ . Therefore,

$$f(\overline{x}) \leq O(Rm).$$

Now, using this bound on $f(\overline{x})$ and $\overline{x}$ as a starting solution for Algorithm 2, we get our result by applying Theorem 4.6. $\qquad\square$

## C  Energy Lemma

**Lemma C.1.** *Let $\widetilde{\Delta} = \arg\min_{Ax=c} x^\top P^\top R P x$. Then one has for any $r$ and $r'$ such that $r' \leq r$,*

$$\Psi(r') \leq \Psi(r) - \frac{1}{2} \sum_i \left( 1 - \frac{r'_i}{r_i} \right) r_i (P\widetilde{\Delta})_i.$$

*Proof.*

$$\Psi(r) = \min_{Ax=c} x^\top P^\top R P x.$$

Constructing the Lagrangian and noting that strong duality holds,

$$\Psi(r) = \min_x \max_y \quad x^\top P^\top R P x + 2y^\top(c - Ax)$$

$$= \max_y \min_x \quad x^\top P^\top R P x + 2y^\top(c - Ax).$$

Optimality conditions with respect to $x$ give us,

$$2P^\top R P x^\star = 2A^\top y.$$

Substituting this in $\Psi$ gives us,

$$\Psi(r) = \max_y \quad 2y^\top c - y^\top A\left(P^\top R P\right)^{-1} A^\top y.$$

Optimality conditions with respect to $y$ now give us,

$$2c = 2A\left(P^\top R P\right)^{-1} A^\top y^\star,$$

which upon re-substitution gives,

$$\Psi(r) = c^\top \left(A\left(P^\top R P\right)^{-1} A^\top\right)^{-1} c.$$

We also note that

$$x^\star = \left(P^\top R P\right)^{-1} A^\top \left(A\left(P^\top R P\right)^{-1} A^\top\right)^{-1} c. \tag{5}$$

We now want to see what happens when we change $r$. Let $R$ denote the diagonal matrix with entries $r$ and let $R' = R - S$, where $S$ is the diagonal matrix with the changes in the resistances. We will use the following version of the Sherman-Morrison-Woodbury formula multiple times,

$$(X + UCV)^{-1} = X^{-1} - X^{-1}U(C^{-1} + VX^{-1}U)^{-1}VX^{-1}.$$

We begin by applying the above formula for $X = P^\top R P$, $C = -I$, $U = P^\top S^{1/2}$ and $V = S^{1/2} P$. We thus get,

$$\left(P^\top R' P\right)^{-1} = \left(P^\top R P\right)^{-1} + \left(P^\top R P\right)^{-1} P^\top S^{1/2}$$

$$\left(I - S^{1/2} P\left(P^\top R P\right)^{-1} P^\top S^{1/2}\right)^{-1} S^{1/2} P\left(P^\top R P\right)^{-1}. \tag{6}$$

We next observe that,

$$I - S^{1/2} P\left(P^\top R P\right)^{-1} P^\top S^{1/2} \preceq I,$$

which gives us,

$$\left(P^\top R' P\right)^{-1} \succeq \left(P^\top R P\right)^{-1} + \left(P^\top R P\right)^{-1} P^\top S P\left(P^\top R P\right)^{-1}. \tag{7}$$

This further implies,

$$A\left(P^\top R' P\right)^{-1} A^\top \succeq A\left(P^\top R P\right)^{-1} A^\top + A\left(P^\top R P\right)^{-1} P^\top S P\left(P^\top R P\right)^{-1} A^\top. \tag{8}$$

We apply the Sherman-Morrison formula again for, $X = A\left(P^\top R P\right)^{-1} A^\top$, $C = I$, $U = A\left(P^\top R P\right)^{-1} P^\top S^{1/2}$ and $V = S^{1/2} P\left(P^\top R P\right)^{-1} A^\top$. Let us look at the term $C^{-1} + VX^{-1}U$.

$$C^{-1} + VX^{-1}U = I + S^{1/2} P\left(P^\top R P\right)^{-1} A^\top (A\left(P^\top R P\right)^{-1} A^\top)^{-1} A\left(P^\top R P\right)^{-1} P^\top S^{1/2}$$

$$\preceq I + S^{1/2} P\left(P^\top R P\right)^{-1} P^\top S^{1/2}$$

$$\preceq I + S^{1/2} R^{-1} S^{1/2}.$$

Using this, we get,

$$\left(A\left(P^\top R'P\right)^{-1}A^\top\right)^{-1} \preceq X^{-1} - X^{-1}U(I + S^{1/2}R^{-1}S^{1/2})^{-1}VX^{-1},$$

which on multiplying by $c^\top$ and $c$ gives,

$$\Psi(r') \leq \Psi(r) - c^\top X^{-1}U(I + S^{1/2}R^{-1}S^{1/2})^{-1}VX^{-1}c.$$

We note from Equation (5) that $x^\star = \left(P^\top RP\right)^{-1}A^\top X^{-1}c$. We thus have,

$$\Psi(r') \leq \Psi(r) - (x^\star)^\top P^\top S^{1/2}(I + S^{1/2}R^{-1}S^{1/2})^{-1}S^{1/2}Px^\star$$

$$= \Psi(r) - \sum_e (r_e - r'_e)\left(1 + \frac{r_e - r'_e}{r_e}\right)^{-1}(Px^\star)_e$$

$$= \Psi(r) - \sum_e \left(\frac{r_e - r'_e}{2r_e - r'_e}\right)r_e(Px^\star)_e$$

$$\leq \Psi(r) - \frac{1}{2}\sum_e \left(\frac{r_e - r'_e}{r_e}\right)r_e(Px^\star)_e$$

Where the last line follows from the fact $2r_e - r'_e \leq 2r_e$. $\qquad\square$

The next lemma is Lemma C.4 in [ABKS21] which is included here for completeness.

**Lemma C.2.** *Let* $\widetilde{\Delta} = \arg\min_{Ax=c} x^\top P^\top RPx$. *Then one has for any* $r'$ *and* $r$ *such that* $r' \geq r$,

$$\Psi(r') \geq \Psi(r) + \sum_e \left(1 - \frac{r_e}{r'_e}\right)r_e(P\widetilde{\Delta})_e^2.$$

*Proof.*

$$\Psi(r) = \min_{Ax=c} x^\top P^\top RPx.$$

Constructing the Lagrangian and noting that strong duality holds,

$$\Psi(r) = \min_x \max_y \quad x^\top P^\top RPx + 2y^\top(c - Ax)$$

$$= \max_y \min_x \quad x^\top P^\top RPx + 2y^\top(c - Ax).$$

Optimality conditions with respect to $x$ give us,

$$2P^\top RPx^\star = 2A^\top y.$$

Substituting this in $\Psi$ gives us,

$$\Psi(r) = \max_y \quad 2y^\top c - y^\top A\left(P^\top RP\right)^{-1}A^\top y.$$

Optimality conditions with respect to $y$ now give us,

$$2c = 2A\left(P^\top RP\right)^{-1}A^\top y^\star,$$

which upon re-substitution gives,

$$\Psi(r) = c^\top\left(A\left(P^\top RP\right)^{-1}A^\top\right)^{-1}c.$$

We also note that

$$x^\star = \left(P^\top RP\right)^{-1}A^\top\left(A\left(P^\top RP\right)^{-1}A^\top\right)^{-1}c. \tag{9}$$

We now want to see what happens when we change $r$. Let $R$ denote the diagonal matrix with entries $r$ and let $R' = R + S$, where $S$ is the diagonal matrix with the changes in the resistances. We will use the following version of the Sherman-Morrison-Woodbury formula multiple times,

$$(X + UCV)^{-1} = X^{-1} - X^{-1}U(C^{-1} + VX^{-1}U)^{-1}VX^{-1}.$$

We begin by applying the above formula for $X = P^\top RP$, $C = I$, $U = P^\top S^{1/2}$ and $V = S^{1/2}P$. We thus get,

$$\left(P^\top R'P\right)^{-1} = \left(P^\top RP\right)^{-1} - \left(P^\top RP\right)^{-1}P^\top S^{1/2}$$
$$\left(I + S^{1/2}P\left(P^\top RP\right)^{-1}P^\top S^{1/2}\right)^{-1}S^{1/2}P\left(P^\top RP\right)^{-1}. \quad (10)$$

We next claim that

$$I + S^{1/2}P\left(P^\top RP\right)^{-1}P^\top S^{1/2} \preceq I + S^{1/2}R^{-1}S^{1/2},$$

which gives us,

$$\left(P^\top R'P\right)^{-1} \preceq \left(P^\top RP\right)^{-1} -$$
$$\left(P^\top RP\right)^{-1}P^\top S^{1/2}(I + S^{1/2}R^{-1}S^{1/2})^{-1}S^{1/2}P\left(P^\top RP\right)^{-1}. \quad (11)$$

This further implies,

$$A\left(P^\top R'P\right)^{-1}A^\top \preceq A\left(P^\top RP\right)^{-1}A^\top -$$
$$A\left(P^\top RP\right)^{-1}P^\top S^{1/2}(I + S^{1/2}R^{-1}S^{1/2})^{-1}S^{1/2}P\left(P^\top RP\right)^{-1}A^\top. \quad (12)$$

We apply the Sherman-Morrison formula again for, $X = A\left(P^\top RP\right)^{-1}A^\top$, $C = -(I + S^{1/2}R^{-1}S^{1/2})^{-1}$, $U = A\left(P^\top RP\right)^{-1}P^\top S^{1/2}$ and $V = S^{1/2}P\left(P^\top RP\right)^{-1}A^\top$. Let us look at the term $C^{-1} + VX^{-1}U$.

$$-\left(C^{-1} + VX^{-1}U\right)^{-1} = \left(I + S^{1/2}R^{-1}S^{1/2} - VX^{-1}U\right)^{-1} \succeq (I + S^{1/2}R^{-1}S^{1/2})^{-1}.$$

Using this, we get,

$$\left(A\left(P^\top R'P\right)^{-1}A^\top\right)^{-1} \succeq X^{-1} + X^{-1}U(I + S^{1/2}R^{-1}S^{1/2})^{-1}VX^{-1},$$

which on multiplying by $c^\top$ and $c$ gives,

$$\Psi(r') \geq \Psi(r) + c^\top X^{-1}U(I + S^{1/2}R^{-1}S^{1/2})^{-1}VX^{-1}c.$$

We note from Equation (9) that $x^\star = \left(P^\top RP\right)^{-1}A^\top X^{-1}c$. We thus have,

$$\Psi(r') \geq \Psi(r) + (x^\star)^\top P^\top S^{1/2}(I + S^{1/2}R^{-1}S^{1/2})^{-1}S^{1/2}Px^\star$$
$$= \Psi(r) + \sum_e \left(\frac{r'_e - r_e}{r'_e}\right)r_e(Px^\star)_e.$$

$\square$