# OpenReview forum: "Unifying Width-Reduced Methods for Quasi-Self-Concordant Optimization"
_NeurIPS.cc/2021/Conference — NeurIPS 2021 Poster_

### Official Review · Reviewer_JKQc · 2021-07-16

**Rating:** 6
**Confidence:** 4

**Summary:**

The authors give an algorithm for solving optimization problems of the form $\min_{x \in \mathbb{R}^m, Ax =b } \sum_i f( (Px)_i)$, where $A,b,P$ are given and $f$ is convex and $\textit{quasi-self concordant}$. This property, introduced by Bach in [Bac10], is an alternative notion to the standard notion of self-concordance and implies that the Hessian of f(x) is multiplicatively stable inside of a box. As several standard regression objectives (softmax, logistic loss, l_p norms) are quasi-self concordant, the study of algorithms for such objective functions is of obvious theoretical and practical importance. The previous state of the art algorithm by [CJJ+20] solves qSC regression problems in $O(m^{1/3})$ linear system solves-- this improves upon previous work of [Bac10] which achieved a complexity of $O(m^{1/2})$ . However, [CJJ+20]'s algorithm (which combines an accelerated gradient method of [MS13] with an efficient algorithm to solve trust region problems) is fairly complex and inherently requires multiple iterative loops.

The present work improves upon [CJJ+20] by providing a simple $O(m^{1/3})$-iteration algorithm for this problem, based on a framework of [CKMST11] and [EV19].


**Limitations And Societal Impact:**

Although the algorithm obtains an alternative to [CJJ+20] which matches the $O(m^{1/3})$ iteration complexity for quasi-self concordant optimization, its theoretical guarantee is still inferior even for the restricted case of qSC functions considered in this paper. For the important special case of softmax regression, the algorithm here obtains a $(1+\nu)$-approximate solution in $O(m^{1/3} \nu^{-5/3})$ iterations-- the result of [CJJ+20] uses $O(m^{1/3} \nu^{-2/3})$ iterations. This gap is surprising, as the algorithm of [EV19] (which the paper borrows heavily from) achieves  $O(m^{1/3} \nu^{-2/3})$ iterations for the related problem of $(1+\nu)$-approximate $\ell_\infty$ regression. Some discussion of why this gap exists (or an improved analysis which closes this gap) would strengthen the result significantly.

In addition, the algorithm of [CJJ+20] actually achieves a stronger result: the iteration complexity improves if the $2$-norm of the optimal solution $x^\star$ is small. Does the algorithm given here achieve a similar result?

Finally, the algorithm given more than superficially resembles the IRLS procedure of [EV19]. I would appreciate some discussion of the ideas borrowed from previous work in the introduction.

**Main Review:**

This analysis is well written. and extends the scope of problems iterative refinement-least squares approaches can solve significantly (previous work on this area has focused on $\ell_p$-norm, softmax, and logistic regression) to the set of all qSC functions. The authors do this by a clever extension of previous techniques-- this allows them to incorporate second derivative information about the function $f$ in the least-squares problems they solve in each iteration. Further, the approach suggested here may achieve significant practical performance gains over previous work (IRLS approaches have been very successfully used for various regression tasks in practice, and converge much faster than the theoretical iteration bounds would indicate).

**Time Spent Reviewing:**

1

---

> ### Author Response · Authors · 2021-08-10
> **Response to Reviewer JKQc**
>
> We thank the reviewer for their positive and very insightful comments, and we hope the following responses will address their concerns.
>
> **Comparison with EV'19, sub-optimal $\epsilon$ dependence, and ideas borrowed from previous works**
>
> While at a surface level our algorithm resembles the approach of EV'19 [1], when comparing the two algorithms more closely, we note an important difference in how the weights/resistances are updated for a *flow* step. In particular, in order to generalize to q.s.c. functions, our algorithm’s weight updates for a flow step are more in line with (and were, in fact, inspired by) AKPS’19 [2]. Though the number of width reduction steps coincides for our method and EV'19 (wherein both achieve a matching $\epsilon^{-2/3}$ dependence), it remains an open problem how to bring the number of flow steps to coincide with the IRLS perspective, and this appears to be challenging due to some problem-specific tools in the analysis of EV'19 that do not generalize as naturally. We will clarify this in a revised version of the paper.
>
> As mentioned both above and in Section 1.2 of our paper, our overall analysis builds on the algorithms of CKMST'10 [3] and AKPS'19 [2]. More specifically the width step analysis is inspired by CKMST'10, while our approach for the flow step analysis is influenced by AKPS'19. That said, we agree that the influence of these previous works could be made even more explicit, and we will certainly expand our discussion of this topic.
>
>
>
> **Iteration complexity in $\ell_2$ and $\ell_{\infty}$ norms**
>
> This is correct, as we do not know if the algorithm in our paper achieves a bound similar to [CJJ+20] in terms of $||x^*||_2$. In the worst case, we can have $||x^*||_2 =\sqrt{m}||x^*||_\infty$ which results in both convergence rates being essentially the same in terms of the $m$ dependence. It is an interesting open problem to determine if our algorithm can achieve convergence bounds in terms of $||x^*||_2$.
>
> **References**
>
> [1] Alina Ene and Adrian Vladu. "Improved Convergence for $\ell_1 $ and $\ell_{\infty} $ Regression via Iteratively Reweighted Least Squares." In International Conference on Machine Learning, pp. 1794-1801. PMLR, 2019.
>
> [2] Deeksha Adil, Rasmus Kyng, Richard Peng, and Sushant Sachdeva. "Iterative refinement for $\ell_p$-norm regression." In Proceedings of the Thirtieth Annual ACM-SIAM Symposium on Discrete Algorithms, pp. 1405-1424. Society for Industrial and Applied Mathematics, 2019.
>
> [3] Paul Christiano, Jonathan A. Kelner, Aleksander Madry, Daniel A. Spielman, and Shang-Hua Teng. "Electrical flows, laplacian systems, and faster approximation of maximum flow in undirected graphs." In Proceedings of the Forty-Third Annual ACM Symposium on Theory of Computing, pp. 273-282. 2011.

---

### Official Review · Reviewer_NMEA · 2021-07-19

**Rating:** 6
**Confidence:** 1

**Summary:**

Width-reduced methods have been used to solve specific optimization problems like maximum folw and lp regression with a small number of iteration $O(m^{1/3})$. However, the design of such methods is tedious because of the need to design specific potentials and analyses. This work aims to give a unified analysis of these methods in the case of quasi-self-concordant functions. This general setting allows to consider new applications like logistic regression.

**Limitations And Societal Impact:**

-

**Main Review:**

This paper is quite technical and assume a good knowledge of previous works. The theoretical results are well stated and seems strong. However, I am not familiar enough with width-reduced methods to be able to check the results.

The quality of the writing is good but I found a lack of informal explanations and motivations of the setting. I am certain to find them in the references given but it would be great to recall them. I hope the following remarks will help to improve the clarity of the paper to non-experts.

- What is the main idea of reduced methods in a simple setting like maximum flow ? In this abstract setting is not even obvious why it is called "with-reduced".
- Why this method first developed for maximum flow is suitable for other settings which seems quite different like logistic regression ?
- What is the intuition behind those potential functions ?
- The use of big O in algorithms is a bit weird. Can't you give a precise value for tau and alpha ?
- I understand that the motivation to go from $O(m^{1/2})$ to $O(m^{1/3})$ in terms of number of call to a linear solver is important in some problems like maximum flow. However, it is quite unusual in logistic regression for example. Is it still a good way to measure computational complexity ? Could you compare to other work considering logistic regression optimization like (Marteau-Ferey 2019) ?

**Time Spent Reviewing:**

5

---

> ### Author Response · Authors · 2021-08-10
> **Response to Reviewer NMEA**
>
> We thank the reviewer for their positive comments, and we hope the following will address their concerns.
>
> > What is the main idea of reduced methods in a simple setting like maximum flow? In this abstract setting is not even obvious why it is called "with-reduced".
>
> The notion of width is common in the *multiplicative weights* literature [1, 2, 3]. Most of these algorithms repeatedly solve a certain subproblem, and “width” is defined as an upper bound on the $\ell_{\infty}$-norm of the solution to these subproblems. In our algorithm, the subproblem is minimizing a weighted $\ell_2$-norm objective (Line 8 of Algorithm 1). The runtime of such algorithms depends linearly on the width, and since this quantity can have a large value, several width-reduction approaches have been proposed that are tailored to specific problems [1,2,3,4,5]. Drawing a parallel to the multiplicative weights algorithms, without a width reduction procedure (Lines 14-18 of Algorithm 1) our algorithm would achieve a $m^{1/2}$ rate of convergence, which is the width of our subproblem. Width reduction allows us to reduce this to $m^{1/3}$ iterations in a way that works more generally than previous problem-specific analyses.
>
> > Why this method first developed for maximum flow is suitable for other settings which seems quite different like logistic regression?
>
> The maximum flow problem can equivalently be expressed as a constrained $\ell_{\infty}$-regression problem, which can be closely approximated by $softmax_{\mu}$ for small $\mu$. The work of CKSMT'10 [4] works with the softmax formulation. The key insight of our work is a width reduction procedure that applies to the more general setting of *q.s.c. functions* which includes both softmax and logistic regression, among others.
>
> > What is the intuition behind those potential functions?
>
> In most acceleration algorithms, at every point the function has well-behaved upper and lower bounds. We analyze the change in the difference between these bounds over iterates, which gives us the runtime of the algorithm. Our algorithm does not follow this exact scheme of acceleration, but it does draw inspiration from it, whereby we can think of $\Phi$ as an upper bound to the function. Specifically, in the case of $\ell_p$-regression it is apparent why this is an upper bound (refer to AKPS’19 [5]). For the q.s.c. case, we observe in Theorem 3.3 that even here we use $\Phi$ to bound the final function value, and such a bound holds for all iterates (not just the final value). The other potential, $\Psi$, is a lower bound on $\Phi$ (not the function) as observed in Lemma 2.4. Also note from the same lemma that $\Psi$ is not too much smaller than $\Phi$. Analogous to acceleration schemes, the key here is to carefully analyze the change in the difference in these two potentials, and that gives us our runtime bounds.
>
> > The use of big O in algorithms is a bit weird. Can't you give a precise value for tau and alpha?
>
> We are using $\tilde{O}$ in the paper to hide absolute constants and logarithmic factors (in problem dependent parameters), for improved readability. We will include the precise values for completeness in the updated version.
>
> > I understand that the motivation to go from $O(m^{1/2})$ to $O(m^{1/3})$ in terms of number of call to a linear solver is important in some problems like maximum flow. However, it is quite unusual in logistic regression for example. Is it still a good way to measure computational complexity? Could you compare to other work considering logistic regression optimization like (Marteau-Ferey 2019)?
>
> Since unregularized logistic regression is not strongly convex, first-order methods incur a poly$(1/\epsilon)$ dependence for reaching an $\epsilon$-accurate solution. In contrast, the works that achieve high-accuracy ($\log(1/\epsilon)$) rates (e.g., Karimireddy et al. (2018) [6], Marteau-Ferey et al. (2019) [7], CJJ+20 [8]) do so by reducing the computation to linear system solves. We will further clarify this distinction in our updated discussion of the related works which will include an additional comparison to Marteau-Ferey et al. (2019).
>
> **References**
>
> [1] Plotkin SA, Shmoys DB, Tardos É. Fast approximation algorithms for fractional packing and covering problems. Mathematics of Operations Research. 1995 May;20(2):257-301.
>
> [2] Lisa K. Fleischer: Approximating fractional multicommodity flow independent of the number of commodities. SIAM Journal of Discrete Mathematics, 13(4):505–520, 2000. Preliminary version in FOCS’99
>
> [3] Garg N, Könemann J. Faster and simpler algorithms for multicommodity flow and other fractional packing problems. SIAM Journal on Computing. 2007;37(2):630-52.
>
> [4] Paul Christiano, Jonathan A. Kelner, Aleksander Madry, Daniel A. Spielman, and Shang-Hua Teng. "Electrical flows, laplacian systems, and faster approximation of maximum flow in undirected graphs." In Proceedings of the Forty-Third Annual ACM Symposium on Theory of Computing, pp. 273-282. 2011.
>
> [5] Deeksha Adil, Rasmus Kyng, Richard Peng, and Sushant Sachdeva. "Iterative refinement for $\ell_p$-norm regression." In Proceedings of the Thirtieth Annual ACM-SIAM Symposium on Discrete Algorithms, pp. 1405-1424. Society for Industrial and Applied Mathematics, 2019.
>
> [6] Sai Praneeth Karimireddy, Sebastian U. Stich, and Martin Jaggi. "Global linear convergence of Newton's method without strong-convexity or Lipschitz gradients." arXiv preprint arXiv:1806.00413 (2018).
>
> [7] Ulysse Marteau-Ferey, Francis Bach, and Alessandro Rudi. "Globally Convergent Newton Methods for Ill-conditioned Generalized Self-concordant Losses." Advances in Neural Information Processing Systems 32 (2019).
>
> [8] Yair Carmon, Arun Jambulapati, Qijia Jiang, Yujia Jin, Yin Tat Lee, Aaron Sidford, and Kevin Tian. "Acceleration with a Ball Optimization Oracle." Advances in Neural Information Processing Systems 33 (2020).

---

> > ### Comment · Reviewer_NMEA · 2021-08-31
> > **Thanks for the reply**
> >
> > Thank you for this detailed answer. I understand much better now. However, reading other reviews confirm me in my opinion. It is good work but the paper would benefit from having more informal explanations and more comparisons to other works. I encourage the authors to include in the paper some of the remarks/explanations made in their responses.

---

### Official Review · Reviewer_J1te · 2021-08-14

**Rating:** 6
**Confidence:** 4

**Summary:**

The paper provides novel, simpler algorithms that match (or come close to matching) the best known results for the minimization of linearly constrained problems of the form $\min_{Ax = b} \sum_i f((Px)_i),$ where $f$ is convex, quasi-self-concordant and $\Px^*|_\infty \leq R$ for the optimum solution. The main contribution is the extension of width-reduction techniques, that previously were confined to l_p-regression problems, to this more general class.



**Limitations And Societal Impact:**

The authors should provide a closer comparison to CJJ+20 in terms of running times. The dependence on the norm of the optimum solution appears to be different there.

**Main Review:**

The strength of this paper relies in its originality, i.e., applying width-reduction techniques to a much wider class of problems than previously done, quasi-self-concordant function.. It is nice that this can be done in a unified way. On the other hand, one would hope that a unified presentation of this approach would allow for some simplification and abstraction of the fundamental steps. On the contrary, the paper feels like a lifting of CKM+11 to a much more general setting without much additional understanding of the width-reduction phenomenon.
Specifically, it would be nice if the authors better motivated the significance of the second derivatives of $f''$, which appears to replace the natural generalization of the MWU of CKM+11, which would be $f'$.

More generally, the paper suffers from a lack of clarity that makes it very difficult to approach for readers well-versed in optimization, but not very familiar with precedent results. For instance, I am very familiar with the seminal result of CKM+11, on which the paper approach is based, but had a very hard time reconciling my knowledge with the current paper because of the poor presentation. Specific points that need to be clarified include:
- make the application of width reduction explicit: in the current version of the paper, the authors do not even mention which variables are performing MWU (presumably the $f''(w)$) and which width is being reduced (that of the flow P\Delta). This clarification would immediately motivate the introduction of the potential function $\Phi.$
- the dual potential CKM+11 is also poorly motivated and should be directly related to typical dual quantities, just as was done in CKM+11. While this is implicitly done in Lemma 2.4, its current presentation remains fairly obscure.

Minor Issues:
- Theorem 5.1 and line 53 appear to use two different notions of the variable $\epsilon$
- line 134: clarify the introduction of the coordinate-wise monotonic property for $f''$.


**Time Spent Reviewing:**

3

---

> ### Author Response · Authors · 2021-08-20
> **Response to Reviewer J1te**
>
> We thank the reviewer for their positive comments, and we hope the following will address their concerns.
>
> >Specifically, it would be nice if the authors better motivated the significance of the second derivatives of f″, which appears to replace the natural generalization of the MWU of CKM+11, which would be f′.
>
> We think that the right generalization is f’’ rather than $f’$ for the following reasons:
>
> - Consider the Iteratively Reweighted Least Squares (IRLS) algorithm for $\ell_p$-regression. The idea behind this algorithm is solving a fixed point problem $||x^*||_p^p = min_x \sum_i |x^*_i|^{p-2}x_i^2$. Note that here the linear system we need to solve involves f’’ (modulo $O(p)$ factors). Since each step of our algorithm also requires solving a quadratic system, our algorithm is similar to IRLS algorithms.
>
> - Another instance where we see the use of f’’ is in the width reduced solver of AKPS’19[1] (Algorithm 4). In AKPS’19[1], their goal is to minimize a smoothed $\ell_p$-norm objective which is $ \ell_2^2 + \ell_p^p$ for $p\geq 2$. This function is also q.s.c., and their algorithm uses f’’ (modulo $O(p)$ factors) for the linear system solver.
> - A third connection is by noting that in AKPS’19[1] (Lemma 5.7), the analysis bounds the second order term using the quadratic subproblem, and the first order term (gradient term) is bounded using Cauchy-Schwarz along with the bound on the second order term. The analogue of this in CKM+’11 is equation (6).
>
> We thus believe that for generalizing CKM+’11 to q.s.c. functions we require f’’.
>
> >make the application of width reduction explicit [...] This clarification would immediately motivate the introduction of the potential function $\Phi$.
>
> In our algorithm, as rightly pointed by the reviewer, the width of $|P\Delta|$ is being reduced. The weight updates, however, are not entirely multiplicative. For a width step it is multiplicative in f’’(w) (lines 14-18 of Algorithm 1), but for a flow step (line 11, Algorithm 1) we perform a purely additive update directly on the weights. We will clarify this in our final version.
>
> >the dual potential CKM+11 [...] should be directly related to typical dual quantities, just as was done in CKM+11.
>
> If we correctly understand the reviewer's comment, we would like to note the following intuitive explanation for the potentials (as also mentioned in our response to Reviewer NMEA):
>
>  When working in the q.s.c. setting, we observe in Theorem 3.3 that here we use the $\Phi$ potential to bound the final function value, and such a bound holds for all iterates (not just the final value). The other potential, $\Psi$, is a lower bound on $\Phi$, as observed in Lemma 2.4, and we may also note from the same lemma that $\Psi$ is not too much smaller than $\Phi$. The key here is to carefully analyze the change in the difference in these two potentials, which ultimately gives us our runtime bounds. We will further elaborate on this motivation in our updated version.
>
> >The authors should provide a closer comparison to CJJ+20 in terms of running times. The dependence on the norm of the optimum solution appears to be different there.
>
> As we noted in our response to Reviewer JKQc, it is indeed the case that CJJ+20 depends on $||x^*||_2$, and we do not know if our algorithms can achieve a similar dependence. However, we have $||x^*||_2 \leq m^{1/2} ||x^*||_\infty$,  and the inequality is tight in the worst case, giving identical iteration counts for the algorithms in terms of m and $||x^*||$. We will include a careful comparison in the updated version.
>
> References:
>
> [1] Deeksha Adil, Rasmus Kyng, Richard Peng, and Sushant Sachdeva. "Iterative refinement for $\ell_p$-norm regression." In Proceedings of the Thirtieth Annual ACM-SIAM Symposium on Discrete Algorithms, pp. 1405-1424. Society for Industrial and Applied Mathematics, 2019.

---

### Official Review · Reviewer_wfoS · 2021-08-16

**Rating:** 6
**Confidence:** 4

**Summary:**

This paper presents a framework for exactly solving optimization problems of the form $\min_{Ax=b} \sum_{i=1}^m f(x_i)$, where $f$ is a 1-dimensional quasi-self-concordant convex function, which means that its second derivative doesn't change multiplicatively by more than $O(1)$ in any interval of size $1$. Examples of quasi-self-concordant functions include the logistic loss, softmax, and $\ell_p$ regression.

The main contribution of the paper is a general algorithm for this problem that works by solving a sequence of $\widetilde{O}(m^{1/3} R \log \frac{1}{\epsilon})$ linear systems where $R = \left\Vert x^*\right\Vert_\infty$. It is based on approximately optimizing a series of $\ell_{\infty}$-constrained quadratic objectives, which is done by a width-reduced multiplicative weights approach [CKM+11] that has been previously been used for max flow, $\ell_p$ regression, and matrix scaling.

**Limitations And Societal Impact:**

-

**Main Review:**

The problem studied is fundamental and important, as can be witnessed by the fact that it contains logistic regression as a special case. On the other hand, the fact that it has an $O(m^{1/3})$ runtime overhead might make it less intriguing to the machine learning community. This could be mitigated if it were accompanied by numerical experiments that would show that this $m^{1/3}$ doesn't arise in practice, but currently there is no evidence that this is the case.

While the general ideas presented are not new and have arisen in previous work, the authors aim to unify all of these approaches by presenting a general all-encompassing analysis. This is a worthy goal, although currently I cannot say that the writing of the paper is clear enough to warrant the status of a go-to reference for the width-reduction approach. In particular, the intuition (both high-level and technical) behind the analysis could be explained much better. In this paper a clearer comparison to previous work, with clean explanations and proofs would go a long way.

Specific comments:

* I would expect to have seen a comparison of the authors' analysis to that of [AZLOW17] for matrix scaling. In fact, it seems to me that the approach is exactly the same: take $\ell_\infty$-constrained steps by using width-reduced multiplicative weights update. What are the technical differences (e.g. significant simplifications in the analysis)? The authors should definitely elaborate on this.

* For clarity, the use of the potentials $\Phi$ and $\Psi$ should be explained. Also, what do their changes mean for the algorithm?

* Page 6 contains a number of technical lemmas but no explanation on each of them.

* The objective function in Section 5.2 does not seem quasi-self-concordant to me. The second derivative is $p(p-1) (Px)^{p-2}$ which can be as much as $p(p-1) R^{p-2}$. As the problem is constrained, the $\ell_2$ regularization term doesn't necessarily help in making $R$ small (e.g. I can scale down $A$ and $b$ in the linear system $Ax=b$ to force $R$ to be arbitrarily large)

* For $\ell_\infty$ regression, how does the authors' approach compare to that of e.g. [EV19], who obtain a better error dependence?

**Time Spent Reviewing:**

2

---

> ### Author Response · Authors · 2021-08-20
> **Response to Reviewer wfoS**
>
> We thank the reviewer for their positive comments, and we hope the following will address their concerns.
>
> >On the other hand, the fact that it has an $O(m^{1/3})$ runtime overhead might make it less intriguing to the machine learning community. This could be mitigated if it were accompanied by numerical experiments that would show that this $m^{1/3}$ doesn't arise in practice, but currently there is no evidence that this is the case.
>
> The primary focus of our work is a general theoretical framework for faster high-accuracy algorithms for a broad class of functions. There is prior evidence that algorithms similar to the one proposed in the paper have a much better iteration complexity in practice.
>
> - The algorithms by EV’19 for $\ell_1$ and $\ell_{\infty}$ regression are also width reduced algorithms and have an $m^{1/3}$ overhead in the theoretical bounds, but the experimental results presented in Figure 6.1 of the paper suggest a much better dependence on $m$.
>
> - Another instance is APS’19[1], where the authors have proposed an algorithm for $\ell_p$ regression. The algorithm here has a worse overhead of $m^{1/2}$, but the experimental results (Figures 2 and 3) again suggest an almost constant iteration complexity.
>
> We are thus hopeful that our algorithm, or its refinements, will be efficient in practice, and we think it is a good direction for future work.
>
>
> **Comparison with AZLOW’17:**
>
> As noted by the reviewer, Section 4 of our paper is inspired by AZLOW’17. In their paper they focus on soft-max and note that it can be approximated by a quadratic function in an $\ell_{\infty}$ ball of constant radius. In our work, we observe that the radius of the box is related to the q.s.c. parameter and are thus able to generalize their overall framework to all q.s.c. functions. However, our main algorithm in Section 3 is quite different and is more in line with CKM+’11 and AKPS’19[3].
>
>
> **Use of Potentials $\Phi$ and $\Psi$ and page 6 lemmas:**
>
> As we have noted in our response to Reviewers NMEA and J1te, and we include here again for completeness, the following intuitive explanation for the potentials:
>
> When working in the q.s.c. setting, we observe in Theorem 3.3 that here we use the $\Phi$ potential to bound the final function value, and such a bound holds for all iterates (not just the final value). The other potential, $\Psi$, is a lower bound on $\Phi$, as observed in Lemma 2.4, and we may also note from the same lemma that $\Psi$ is not too much smaller than $\Phi$. The key is to carefully analyze the change in the difference in these two potentials, which ultimately gives us our runtime bounds. We will further elaborate on this motivation in our updated version.
>
> For the lemmas on page 6, we will add details in the revised version.
>
> **Quasi-self-concordance of $\ell_p$ regression:**
>
> The reviewer is right that $||Px||_p^p$ is not q.s.c., and the standard approaches of a *homotopy* on the regularizer may not work. However, as noted in (Lemma 16) CJJ+’20[2], $||Px||_p^p + \mu ||Px||_2^2$ is q.s.c., and we can thus apply our algorithm to this regularized problem to obtain a low accuracy solution for the regularized problem. In order to translate this to a solution of the $\ell_p$-regression problem, we can use the result of APS’19[1] (Lemma 3.2) that says it is sufficient to optimize such regularized problems to a constant approximation. Repeating this a few (logarithmically many) times leads to a high accuracy solution for $\ell_p$-regression.
>
> **Comparison with EV’19:**
>
> As we have noted in our response to Reviewer JKQc, at a high level the algorithms in these two papers appear similar. However, there are subtle differences in both approaches. We include the response for completeness.
>
> While at a surface level our algorithm resembles the approach of EV'19, when comparing the two algorithms more closely, we note an important difference in how the weights/resistances are updated for a *flow* step. In particular, in order to generalize to q.s.c. functions, our algorithm’s weight updates for a flow step are more in line with (and were, in fact, inspired by) AKPS’19[3]. Though the number of width reduction steps coincides for our method and EV'19 (wherein both achieve a matching $\epsilon^{-2/3}$ dependence), it remains an open problem how to bring the number of flow steps to coincide with the IRLS perspective, and this appears to be challenging due to some problem-specific tools in the analysis of EV'19 that do not generalize as naturally. We will clarify this in a revised version of the paper.
>
>
> **References:**
>
> [1] Deeksha Adil, Richard Peng, and Sushant Sachdeva. 2019. Fast, provably convergent IRLS algorithm for p-norm linear regression. Proceedings of the 33rd International Conference on Neural Information Processing Systems.
>
> [2] Yair Carmon, Arun Jambulapati, Qijia Jiang, Yujia Jin, Yin Tat Lee, Aaron Sidford, and Kevin Tian. "Acceleration with a Ball Optimization Oracle." Advances in Neural Information Processing Systems 33 (2020).
>
> [3] Deeksha Adil, Rasmus Kyng, Richard Peng, and Sushant Sachdeva. "Iterative refinement for $\ell_p$-norm regression." In Proceedings of the Thirtieth Annual ACM-SIAM Symposium on Discrete Algorithms, pp. 1405-1424. Society for Industrial and Applied Mathematics, 2019.

---

### Author Response · Authors · 2021-08-27
**Happy to Respond to Any Further Comments/Clarifications**

Dear Reviewers,

We would like to thank you all for your insightful comments and suggestions, all of which will greatly help to improve the clarity of our paper.

We hope that our responses have been helpful for providing more insights into the paper’s contributions. We would be glad to respond to any further comments/clarifications that might allow for a better understanding and evaluation of our work. We also kindly request you to consider stronger support for the paper if your concerns have been addressed.

Thank you.

---

### Decision · Program_Chairs · 2021-09-27

**Decision:**

Accept (Poster)

**Comment:**

The reviewers and area chair agree that the technical contribution of the paper is significant as it yields a simpler algorithm for an important class of problems and extends the applicability of the width-reduced approach of [CKM+11]. The main weakness of the paper is the lack of clear explanations behind the technical analysis and comparison with existing similar methods.  However, the authors have addressed the reviewers' questions and provided some explanation in the rebuttal phase. Assuming these explanations will find their way in the final version, I believe the submission should be accepted.